# Homeostasis at different backgrounds: The roles of overlayed feedback structures in vertebrate photoadaptation

**Jonas V. Grini**, **Melissa Nygård**, **Peter Ruoff** *

Department of Chemistry, Bioscience, and Environmental Engineering, University of Stavanger, Stavanger, Norway

☯ These authors contributed equally to this work.
* peter.ruoff@uis.no

**Data Availability Statement:** All relevant data are within the paper and its Supporting information files.

**Funding:** The author(s) received no specific funding for this work.

## Abstract

We have studied the resetting behavior of eight basic integral controller motifs with respect to different but constant backgrounds. We found that the controllers split symmetrically into two classes: one class, based on derepression of the compensatory flux, leads to more rapid resetting kinetics as backgrounds increase. The other class, which directly activates the compensatory flux, shows a slowing down in the resetting at increased backgrounds. We found a striking analogy between the resetting kinetics of vertebrate photoreceptors and controllers based on derepression, i.e. vertebrate rod or cone cells show decreased sensitivities and accelerated response kinetics as background illuminations increase. The central molecular model of vertebrate photoadaptation consists of an overlay of three negative feedback loops with cytosolic calcium ($Ca_i^{2+}$), cyclic guanosine monophosphate (cGMP) and cyclic nucleotide-gated (CNG) channels as components. While in one of the feedback loops the extrusion of $Ca_i^{2+}$ by potassium-dependent sodium-calcium exchangers (NCKX) can lead to integral control with cGMP as the controlled variable, the expected robust perfect adaptation of cGMP is lost, because of the two other feedback loops. They avoid that $Ca_i^{2+}$ levels become too high and toxic. Looking at psychophysical laws, we found that in all of the above mentioned basic controllers Weber's law is followed when a "just noticeable difference" (threshold) of 1% of the controlled variable's set-point was considered. Applying comparable threshold pulses or steps to the photoadaptation model we find, in agreement with experimental results, that Weber's law is followed for relatively high backgrounds, while Stephens' power law gives a better description when backgrounds are low. Limitations of our photoadaption model, in particular with respect to potassium/sodium homeostasis, are discussed. Finally, we discuss possible implication of background perturbations in biological controllers when compensatory fluxes are based on activation.

**Competing interests:** The authors have declared that no competing interests exist.

## Introduction

In 1929 Walter B. Cannon [1] defined homeostasis as the sum of the physiological processes which keep the steady states in a cell or organism within narrow limits [2]. Since then many facets of homeostatic regulation has been discovered and alternative concept names have been suggested. For example, Mrosovsky [3] argued that the term *rheostasis* would be more appropriate since there is often a change in a defended set-point, for example, the elevated (and controlled) temperature when we are running a fever. He further argues (see [3], ch. 1) that homeostasis has often been equated to a single negative feedback loop. The term *allostasis* [4–6] was introduced to focus on changing environmental conditions, feedforward loops, and on the control mechanisms which deviate from a simple negative feedback loop with a single set-point [5]. With respect to circadian adaptation and anticipation mechanisms Moore-Ede [7] coined the term *predictive homeostasis*. As adaptation mechanisms are highly dynamic Lloyd [8] argued for the use of the term *homeodynamics* instead of homeostasis. While all these aspects point to important properties of homeostatic regulation, we agree with Carpenter that the term *homeostasis* still stands as an unified approach [9]. We believe, that when multiple feedback and feedforward loops are studied theoretically in more detail, many of the above mentioned homeostatic facets can be accounted for, such as rheostatic control can be observed in a model of p53 regulation upon variable stress conditions [10].

In this paper we explore the influence of background perturbations on a set of eight basic negative feedback (controller) motifs [11]. We found that some of the motifs show an astonishing analogy to retinal photoreceptor adaptation when various background illuminations are applied.

The paper consists of two major parts. In the first part results from a systematic study of all eight controller motifs are shown. In the second part we show how certain of these controller motifs can provide an understanding about the kinetics of retinal photoreceptor adaptation. All eight feedback motifs show robust perfect homeostasis due to the implementation of integral control.

Integral control is a control-engineering concept [12], which allows a controlled variable to reset precisely at its set-point when step perturbations are applied. In biochemical systems several kinetic requirements have been identified which lead to integral control. Among them we have zero-order kinetics in the removal of the manipulated (controller) variable [11, 13], antithetic control in which two controller variables are removed by second-order [14, 15] or enzyme [16] kinetics, or a (first-order) autocatalytic synthesis combined with first-order removal kinetics of the manipulated variable [17–19]. When dealing with the different basic controller motifs we will introduce integral control mostly by zero-order kinetics, but also by antithetic control (see 'Results and discussion' below).

### Psychophysical laws

Psychophysical laws relate the intensity of a physical stimulus with its perceived magnitude, for example a human (or a receptor cell) perceived brightness of light in relation to a certain applied light intensity. We will use the concept of a "just noticeable perturbation" (alternatively "just noticeable difference" or "threshold") in order to compare computational results with corresponding experimental data. The concept of a "just noticeable difference" was first introduced by Weber [20] in order to understand the relationship between an applied physical stimulus and its (human) perception. We will focus on two well-known psychophysical laws, i.e. on Weber's law and on Stephens' power law, because these laws are often applied in adaptation studies (see for example Part IV in [21])).

**Weber's law.** Ernst Heinrich Weber [20, 22] found that the human perception of a just noticeable difference $dw = w' - w$ between a reference weight $w$ and a slightly heavier weight $w'$ is approximately proportional to the reference weight $w$, i.e.,

$$dw = w' - w = k \cdot w \tag{1}$$

with $k$ being a constant. Weber's law implies a linear relationship between a just noticeable perturbation (threshold perturbation) and an applied background perturbation. It was Gustav Fechner [23] who made Weber's law public and gave it its name, but expanded the perception of a just noticeable difference $dw$ to $dp = dw/w$ (termed by Fechner as *Contrast*) and stated its logarithmic form, i.e.,

$$dp = \alpha \cdot \frac{dw}{w} \quad \Rightarrow \quad p = \alpha \ln \frac{w}{w_0} + C \tag{2}$$

where $\alpha$ and $C$ are constants. Instead of weight, $w$ can generally be any other stimulus. The logarithmic form of Eq 2 is termed as *Fechner's law*.

**Stevens' power law.** Stevens [24] suggested (and revived) a power-law formulation between the magnitude of a sensation/perception $p$ and its stimulus $s$, i.e.

$$p = k \cdot s^\alpha + p_0 \tag{3}$$

where $k$, $\alpha$, and $p_0$ are constants depending respectively on the units used and the type of stimulation. MacKay [25] suggested a model of perceived intensities by an adaptive "counterbalancing" response mechanism. This "negative feedback" approach enabled MacKay to make connections between the Weber-Fechner law and Stevens' law. In a model of retinal light adaptation we will show that Stephens' power law or Weber's law are followed dependent whether the background perturbation range is either low or high, respectively.

## Materials and methods

### Calculations and parameter estimations

Computations were performed by using LSODE [26], which is part of a set of Fortran solvers at the Lawrence Livermore National Laboratory (https://computing.llnl.gov/projects/odepack). Graphical results were generated with gnuplot (www.gnuplot.info). Composite figures and additional annotations were done with Adobe Illustrator (https://www.adobe.com/).

To make notations simpler, concentrations of compounds are denoted by compound names without square brackets. Time derivatives are generally indicated by the 'dot' notation. For the basic feedback loops m1-m8 (next section) concentrations and rate parameter values are given in arbitrary units (au), while for the light adaptation model concentrations are in $\mu$M (or nM) and time scale is in seconds (s). Rate parameters are presented as $k_i$'s ($i = 1, 2, 3, \ldots$) irrespective of their kinetic nature, i.e. whether they represent turnover numbers, Michaelis constants, or inhibition/activation constants.

For the light adaptation model some parameter values were estimated by using gnuplot's fit function with respect to experimental literature data. Graphical experimental data were extracted with the program GraphClick (https://macdownload.informer.com/graphclick/).

To make the computations more accessible supporting information 'S1 Programs' in S1 File contains python scripts of Fortran results.

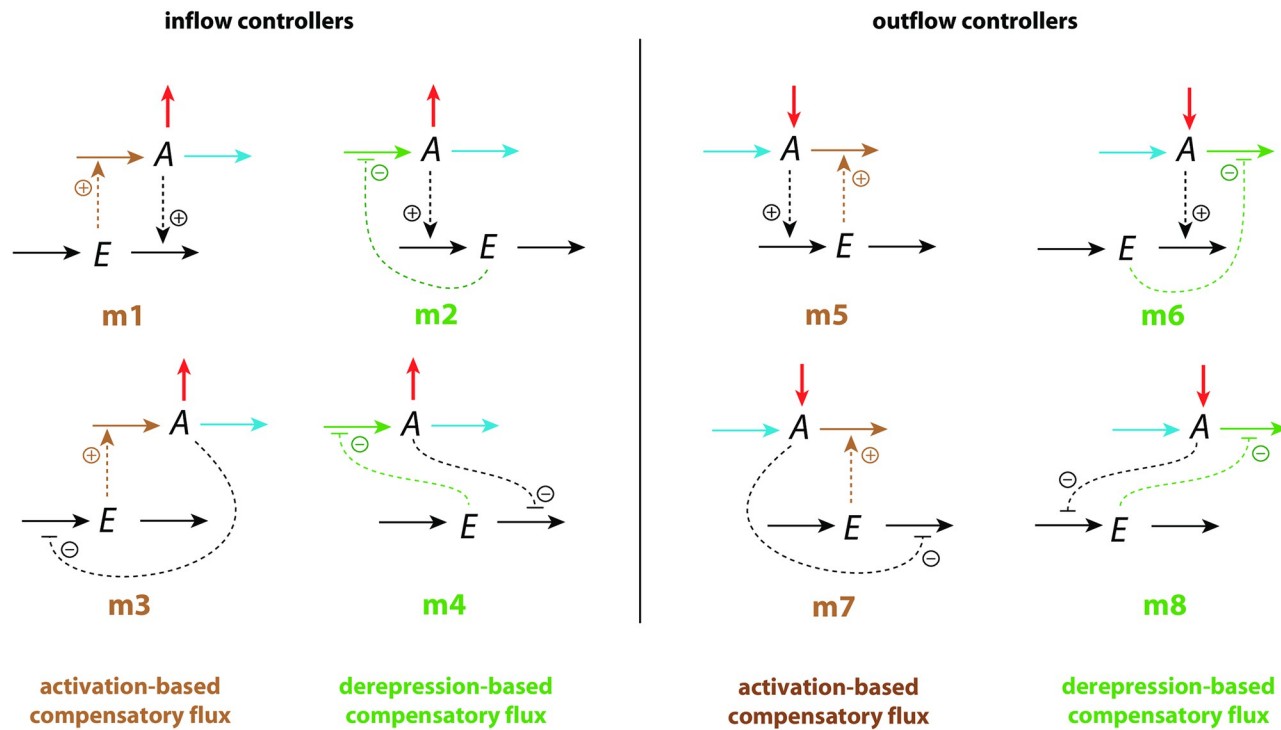

**Fig 1. Set of basic negative feedback motifs m1-m8.** Red and blue arrows indicate, respectively, a step perturbation and a constant background reaction. Integral control is implemented either by zero-order kinetics [11, 13] or by antithetic control [14, 16]. Outlined in brown and green we have activating or derepressing compensatory fluxes, respectively.

### Feedback motifs investigated

Fig 1 shows the investigated negative feedback loops. These are eight basic motifs (m1-m8), which divide equally into a set of inflow and outflow controllers [11]. Compound $A$ is the homeostatic controlled variable, while $E$ is the controller variable (or manipulated variable). Red arrows indicate a step perturbation while blue arrows represent a constant background. Black arrows indicate synthesis and removal of the controller variable $E$. Dashed lines represent signaling events which lead to the activation (plus signs) or inhibition (minus signs) of target reactions.

We have applied step perturbations, because integral controllers are generally capable to compensate them perfectly (for a proof see ch. 10.3.1 in Ref. [27]). Note however, that some feedback loop kinetics, such as in m2, are capable to oppose even rapidly increasing perturbations, such as hyperbolic growth [28, 29].

In the inflow controllers m1-m4 the manipulated variable $E$ leads to the increase of a compensatory inflow flux either by direct activation (brown plus signs) or by derepression (green minus signs) and thereby opposing the step perturbations which remove $A$ (red arrows). In the outflow controllers m5-m8 the compensatory (outflow) flux compensates step perturbations (red arrows) which increase $A$ [30].

## Results and discussion

### Analyses of controller motifs

We have analyzed the eight controller schemes (Fig 1) with regard to step perturbations at different but constant backgrounds. Fig 2 shows the two idealized responses. In panel (a) the

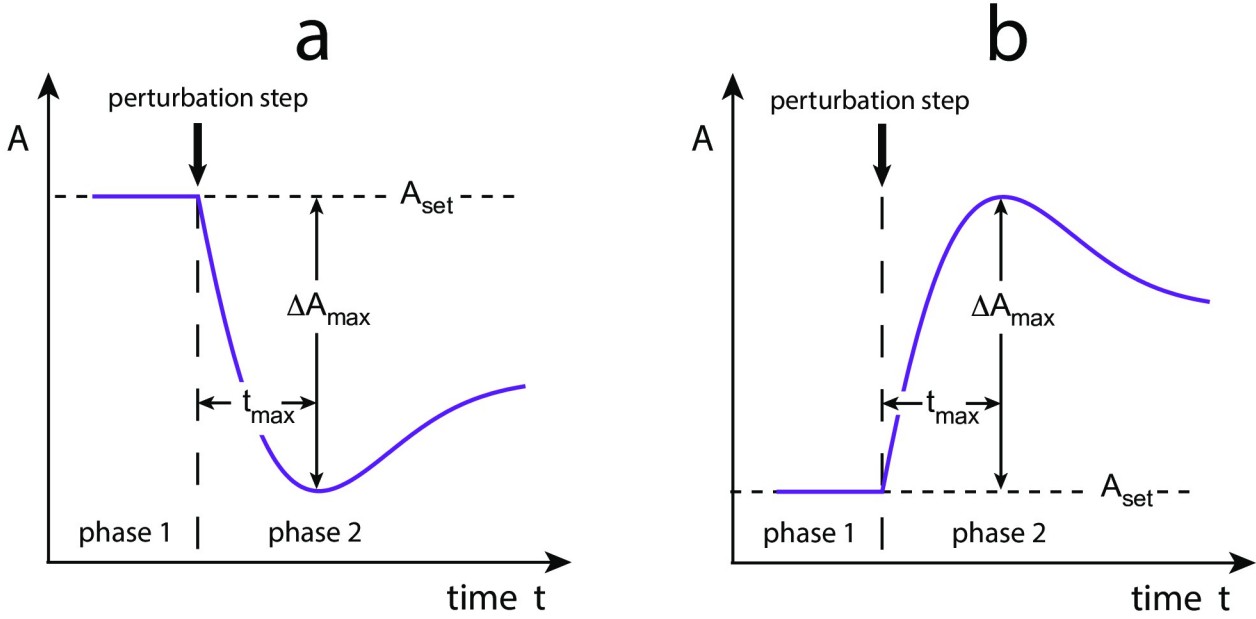

**Fig 2. Idealized response kinetics of (a) inflow and (b) outflow controllers upon step perturbations.** Indicated are the set-point of the controlled variable $A$, $A_{set}$, the maximum excursion of $A$, $\Delta A_{max}$. $t_{max}$ is the time between the start of the perturbation until $\Delta A_{max}$ is reached.

resetting for inflow controllers is shown. In this case a step perturbation removes the controlled variable $A$ and temporarily decreases it. Panel (b) shows the behavior of an outflow controller. When integral control is operative the controllers will defend the set-point of $A$ ($A_{set}$) and move the level of $A$ during the on-going step perturbation precisely back to $A_{set}$.

The *resetting period* is rather loosely defined as the time required to reach $A_{set}$ after a step perturbation has been applied. Fig 2 also indicates the parameter $\Delta A_{max}$, which is the maximum excursion of $A$ after the applied step. $t_{max}$ is the time the controller needs to reach $\Delta A_{max}$ after the perturbation has been applied.

We found that the controllers' response kinetics split into two classes independent whether they are inflow or outflow controllers. In both classes an increase of a background reaction leads to a reduced excursion $\Delta A_{max}$. In the class where the compensatory flux is based on activation (controllers m1, m3, m5, and m7; outlined in brown in Fig 1), the controllers slow down in their resetting with increasing backgrounds and decreasing $t_{max}$ values. In the other class, when compensatory fluxes are based on derepression, the controllers show an accelerated resetting (controllers m2, m4, m6, and m8; outlined in green in Fig 1).

In the following we describe in more detail how the two classes of controllers differ in their resetting behavior.

## Controllers with activated compensatory fluxes

We show here the results for motifs m1 and m7. The supporting information 'S1 Text' shows corresponding details for controllers m3 and m5.

**Controller m1.** In the m1 controller the compensatory flux $j_3 = k_3 \cdot E$ is activated by $E$ while $A$ activates the removal of $E$ (Fig 3). Step-wise perturbations removing $A$ are mediated by $k_2$ while $k_4$ is a constant background outflow. For simplicity, we assume that activation kinetics are first-order with respect to the concentration of the activating species. This

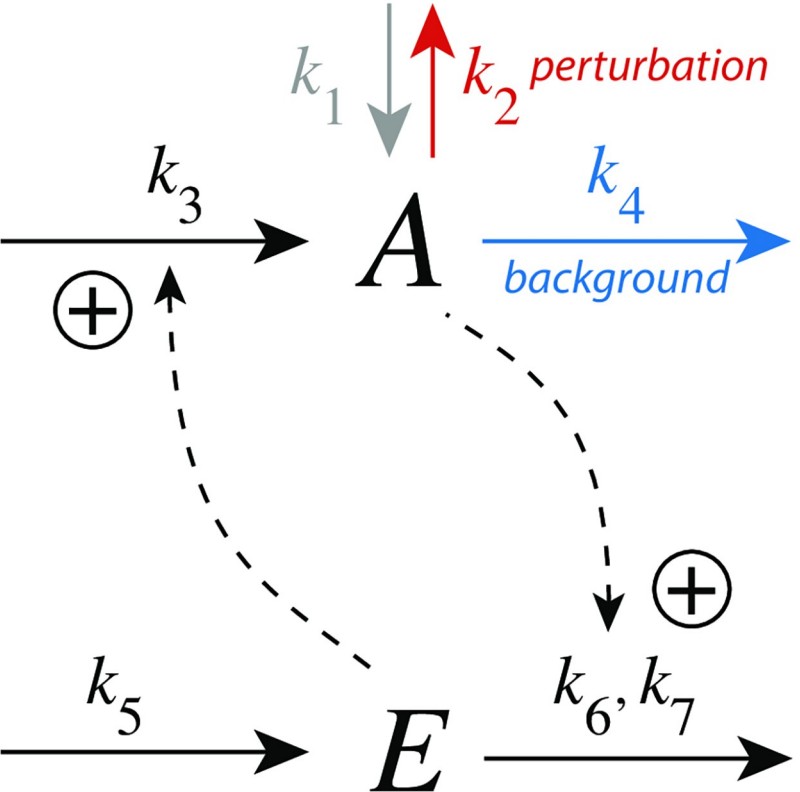

**Fig 3. Inflow controller m1 with integral control implemented as a zero-order Michaelis-Menten (MM) type removal of E.** $k_2$ undergoes a step perturbation, $k_3$ is a rate constant for the inflow of $A$, while $k_4$ is a constant background reaction. $k_6$ and $k_7$ are MM parameters analogous to $V_{max}$ and $K_M$, respectively. In the calculations the grayed-out rate constant $k_1$ will be set to zero.

assumption neglects possible saturation and controller breakdown at high activator concentrations [31].

The rate equations are:

$$\dot{A} = k_1 - (k_2 + k_4) \cdot A + k_3 \cdot E \tag{4}$$

$$\dot{E} = k_5 - A\left(\frac{k_6 \cdot E}{k_7 + E}\right) \tag{5}$$

Integral control is incorporated by a zero-order kinetic removal of $E$, i.e. $E/(k_7 + E) \approx 1$, with the result that $E$ becomes proportional to the integrated error $\epsilon = A_{set} - A$:

$$\dot{E} = k_6 \big( \underbrace{\frac{k_5}{k_6}}_{A_{set}} - A \big) = k_6 \cdot \epsilon \;\; \Rightarrow \;\; E(t) = k_6 \int_0^t \epsilon(t') \cdot dt' \tag{6}$$

Fig 4 shows the response kinetics of the m1 controller with set-point $A_{set}$=3.0. Panel (a) shows the concentration of $A$ as a function of time when a $k_2$ step 1→5 is applied. Clearly, $\Delta A_{max}$ (see definition in Fig 2) decreases with increasing background $k_4$. Typically for controllers where the compensatory flux is based on activation, we observe that for increased backgrounds the

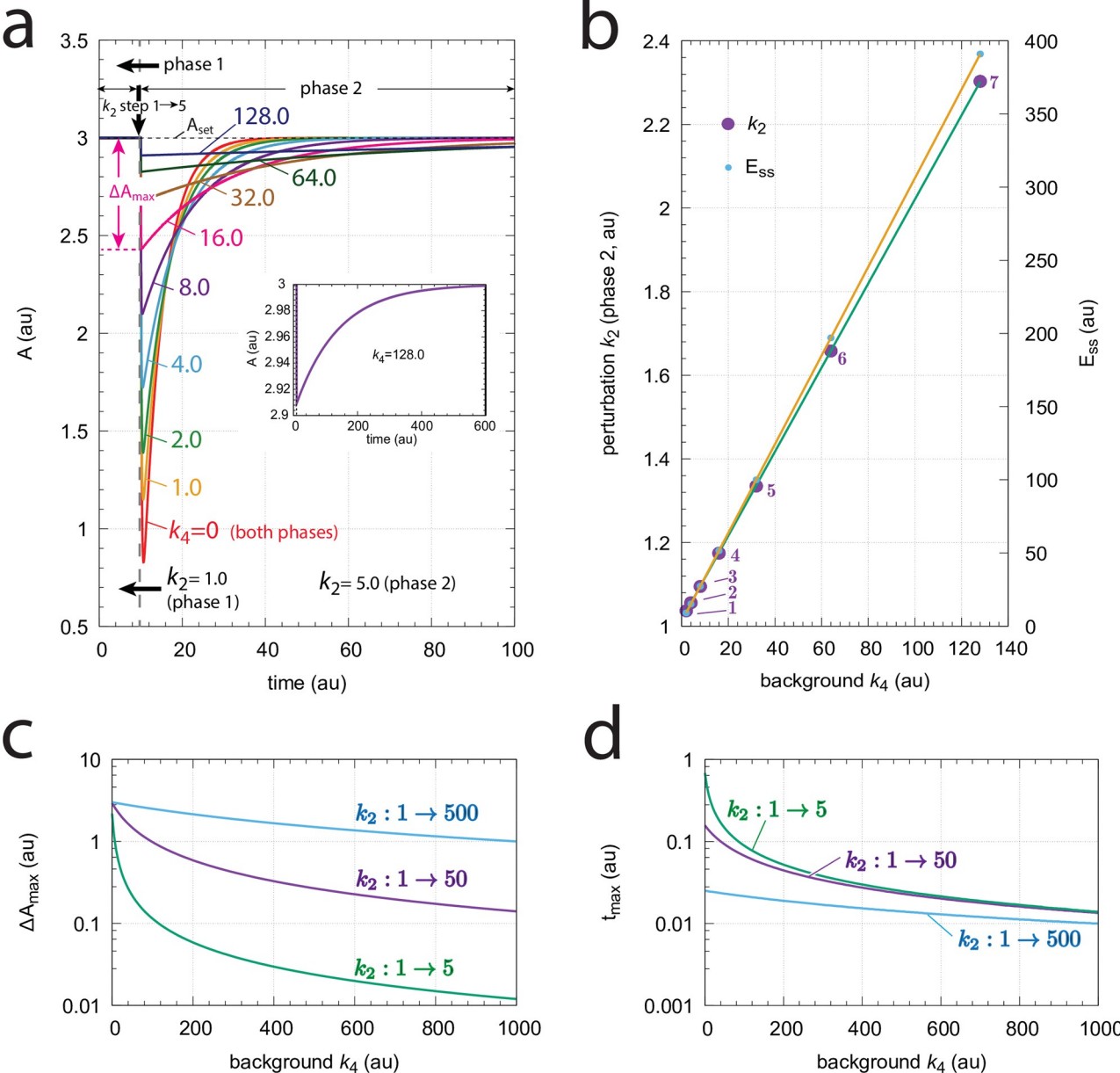

**Fig 4. Response kinetics and relationship to Weber's law in the m1 controller (Fig 3).** The set-point of $A$ is 3.0. (a) Step-wise increase of $k_2$ from 1.0 to 5.0 at time $t$=10 at different but constant backgrounds $k_4$ (0–128.0, phases 1 and 2). Note the successive decrease in the maximum excursion of $A$ ($\Delta A_{max}$) with slowed-down $A$ resetting kinetics as $k_4$ backgrounds increase. $\Delta A_{max}$ for $k_4$=16.0 is indicated. The inset shows that even at high backgrounds the controller is fully operative. Rate constants (in au): $k_1$=0.0, $k_2$=1.0 (phase 1), $k_2$=5.0 (phase 2), $k_3$=1.0, $k_4$ variable, $k_5$=3.0, $k_6$=1.0, $k_7$=1 × 10$^{-6}$. Initial concentrations (in au): $A_0$=3.0, $E_0$=3.0 ($k_4$=0); $A_0$=3.0, $E_0$=6.0 ($k_4$=1); $A_0$=3.0, $E_0$=9.0 ($k_4$=2); $A_0$=3.0, $E_0$=15.0 ($k_4$=4); $A_0$=3.0, $E_0$=27.0 ($k_4$=8); $A_0$=3.0, $E_0$=51.0 ($k_4$=16); $A_0$=3.0, $E_0$=99.0 ($k_4$=32); $A_0$=3.0, $E_0$=195.0 ($k_4$=64); $A_0$=3.0, $E_0$=387.0 ($k_4$=128). The inset shows the full adaptation response when $k_4$=128.0 (b) Relationship to Weber's law: When perturbation $k_2$ in phase 2 is adjusted such that the maximum (just noticeable) excursion in $A$ is 0.03 (i.e. 1% of $A_{set}$) then both $k_2$ and the "perception" $E_{ss}$ are linear functions of different but constant backgrounds $k_4$. Rate constants and initial concentrations as in (a), except that $k_2$ in phase 2 has the following values: **1**, $k_2$ = 1.0367 ($k_4$ = 2); **2**, $k_2$ = 1.0559 ($k_4$ = 4); **3**, $k_2$ = 1.0950 ($k_4$ = 8); **4**, $k_2$ = 1.1745 ($k_4$ = 16); **5**, $k_2$ = 1.3350 ($k_4$ = 32); **6**, $k_2$ = 1.6581 ($k_4$ = 64); **7**, $k_2$ = 2.3030 ($k_4$ = 128). (c) $\Delta A_{max}$ as a function of background $k_4$ at three different $k_2$ steps. (d) $t_{max}$ as a function of background $k_4$ at three different $k_2$ steps. Rate constants are as in panel (a), except for $k_2$ and $k_4$. Initial concentrations are the steady state values of $A$ and $E$ prior to the step in $k_2$.

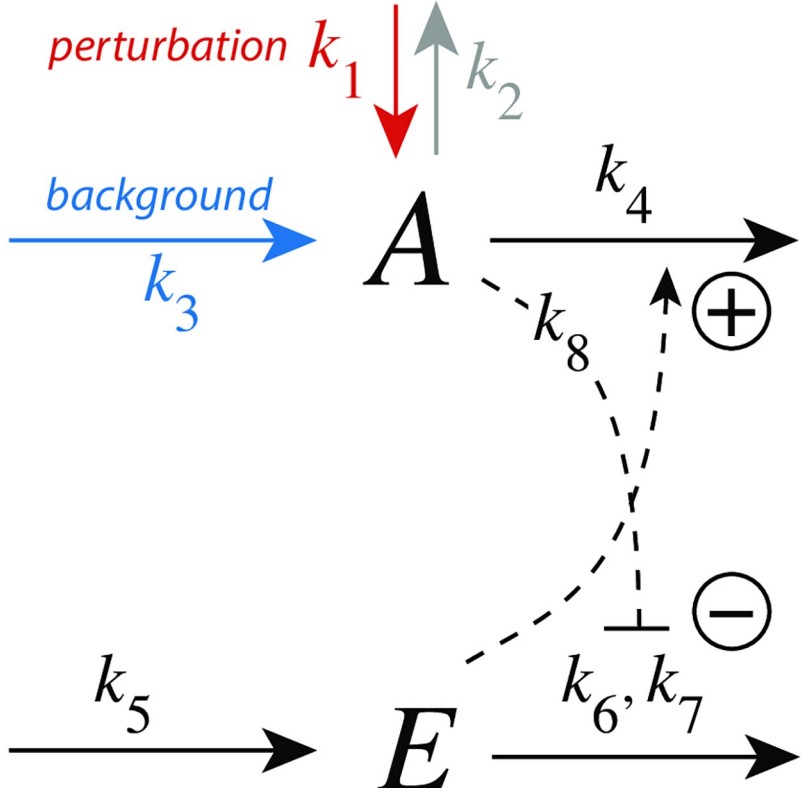

**Fig 5. Outflow controller motif m7 with integral control implemented as a zero-order Michaelis-Menten (MM) type degradation of E.** The perturbation $k_1$ changes step-wise ($1.0 \rightarrow 5.0$), while $k_3$ is a constant background. Rate constant $k_4$ relates to the outflow of $A$, and $k_8$ is an inhibition constant. $k_6$ and $k_7$ are MM parameters analogous to $V_{max}$ and $K_M$, respectively. In the calculations the grayed-out rate constant $k_2$ is, for the sake of simplicity, set to zero.

resetting period is lengthend. Despite the increase in the resetting period the inset in panel (a) shows that the controller is fully operational and is able to defend its set-point.

Fig 4b shows the response kinetics related to Weber's law when probing a "just noticeable" excursion in $\Delta A$ of 0.03 (1% of $A_{set}$=3.0) by applying appropriate $k_2$ values in phase 2. We observe that the different $k_2$ values (in phase 2) and the corresponding steady-state values of $E$ ($E_{ss}$) are linear functions of the background $k_4$.

Fig 4c and 4d show the values of $\Delta A_{max}$ and $t_{max}$ for three different $k_2$ steps with increasing backgrounds $k_4$. Reflecting the behavior from Fig 4a, panel (c) shows that $\Delta A_{max}$ values decrease monotonically as background increases, but that the magnitude of $\Delta A_{max}$ depends on the size of the applied step. Despite that the resetting period increases with increasing backgrounds we observe that $t_{max}$ decreases with increasing $k_4$ (panel (d)). The increase of the resetting period at increased $k_4$ levels can be explained by the high steady state levels of $E$ in phase 1 when $k_4$ backgrounds become high and that the system needs more time to reach the steady state of $E$ in phase 2 by zero-order kinetics.

**Controller m7.** m7 is an outflow controller which opposes inflow perturbations $k_1$ at different background reactions $k_3$ by $E$-activation of the compensatory flux $j_4$ ($=k_4 \cdot A \cdot E$). The negative feedback is closed by inhibiting the removal of $E$ through $A$ (Fig 5). The rate equations are

$$\dot{A} = k_1 + k_3 - k_2 \cdot A - k_4 \cdot A \cdot E \qquad (7)$$

$$\dot{E} = k_5 - \left(\frac{k_6 \cdot E}{k_7 + E}\right) \cdot \left(\frac{k_8}{k_8 + A}\right) \tag{8}$$

The set-point for $A$ is calculated from the steady-state condition of Eq 8 by using zero-order degradation of $E$, i.e. $E/(k_7 + E) \approx 1$.

$$\dot{E} = 0 \;\Rightarrow\; k_5 = \frac{k_6 k_8}{(k_8 + A_{ss})} \;\Rightarrow\; A_{set} = A_{ss} = k_8\left(\frac{k_6}{k_5} - 1\right) \tag{9}$$

Fig 6 shows the response kinetics of the m7 controller. Since the controller opposes inflow perturbations excursions of $A$ are above the set-point $A_{set}$ (=3.0). Panel a shows the slowed-down responses during the resetting in phase 2 as background $k_3$ increases. The inset shows that the controller is still operative even at the highest $k_3$ and slowest resetting. Panel b shows that a $k_1$ step perturbation which results in a just noticeable maximum excursion $\Delta A_{max}$ of 0.03 (1% of $A_{set}$) increases, together with the corresponding steady state $E_{ss}$ values in phase 2, linearly with the background $k_3$. $\Delta A_{max}$ in creases with increasing $k_1$ step (Fig 6c), while for a given background we find, somewhat surprisingly, that $t_{max}$ is independent on the magnitude of the $k_1$ step (Fig 6d). Both $\Delta A_{max}$ and $t_{max}$ decrease monotonically with increasing background $k_3$.

We explain the delay in the resetting of $A$ for large $k_3$ backgrounds as the increased time needed to change the high steady state values of $E$ from phase 1 to its new steady state in phase 2 after the step.

### Controllers with compensatory fluxes based on derepression

We show here the results for controllers m2 and m8 (Fig 1). Corresponding results for m4 and m6 are given in supporting information 'S2 Text'.

**Controller m2.** In the m2 controller scheme (Fig 7) activation of $E$ by $A$ is proportional to the concentration of $A$, while the inhibition term on the compensatory flux is formulated as $k_8/(k_8 + E)$. The rate equations are:

$$\dot{A} = k_1 - k_2 \cdot A - k_4 \cdot A + \frac{k_3 k_8}{k_8 + E} \tag{10}$$

$$\dot{E} = k_5 \cdot A - \frac{k_6 \cdot E}{k_7 + E} \tag{11}$$

To achieve homeostasis in $A$ a perturbation (removal) of $A$ is counteracted by a decrease of $E$ ("derepression"), which increases the compensatory flux $j_3 = k_3 k_8/(k_8 + E)$ and moves, in the presence of integral control, $A$ to its set-point.

The set-point of $A$ ($A_{set}$) is determined how integral control is implemented in the feedback loop. In Fig 7 we use zero-order kinetics with respect to the removal of $E$, i.e. $k_7 \ll E_{ss}$. This implies that the steady state of $A$ is also the set-point of $A$ ($A_{set}$) and is given as the ratio $k_6/k_5$, i.e.

$$\dot{E} = 0 = k_5 \cdot A_{ss} - k_6 \cdot \underbrace{f_E}_{\approx 1} = -k_5\Big(\underbrace{\frac{k_6}{k_5}}_{A_{set}} - A_{ss}\Big) \tag{12}$$

with $f_E = E/(k_7 + E) \approx 1$.

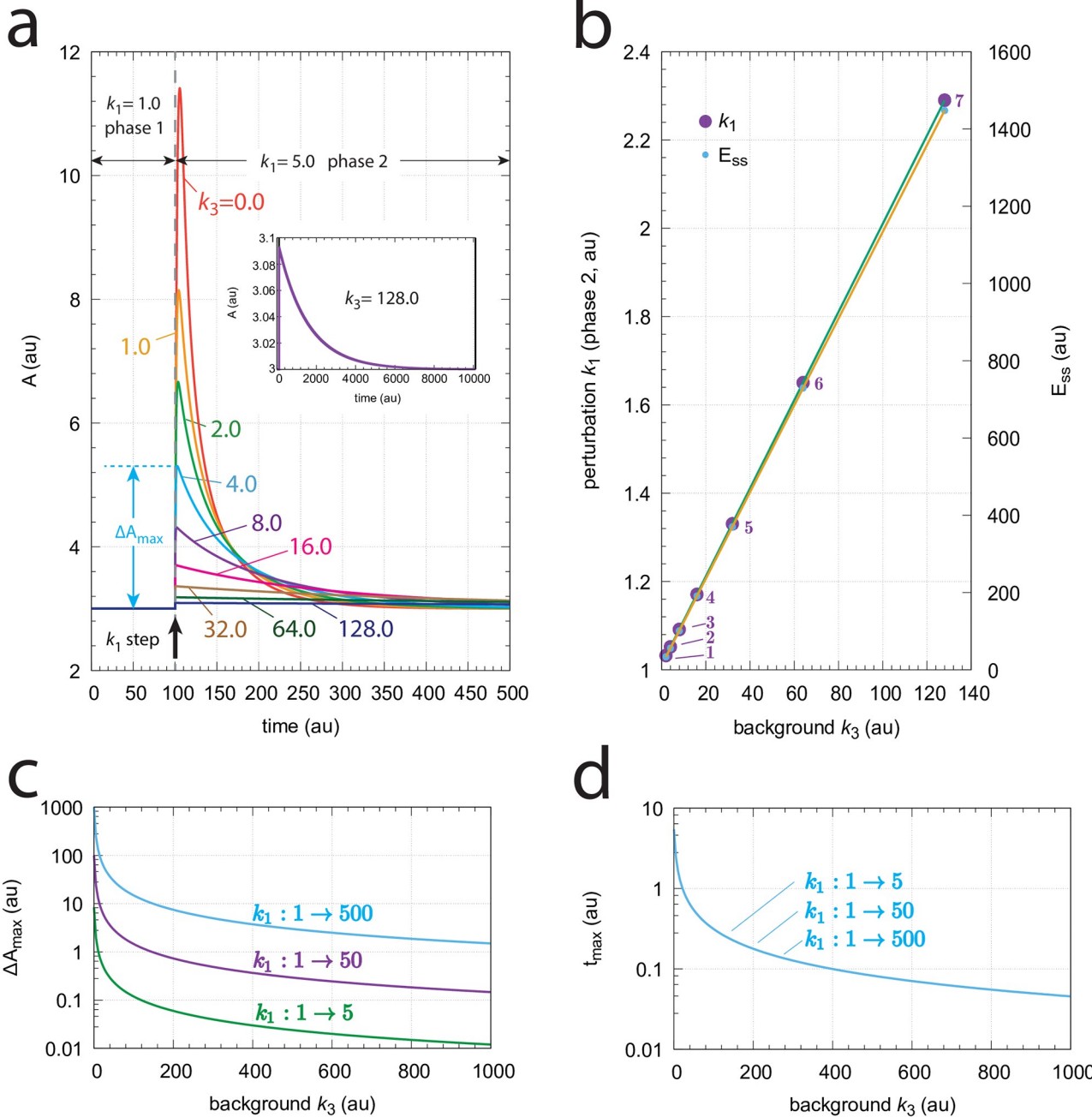

**Fig 6. Response kinetics and relationship to Weber's law in the m7 controller (Fig 5).** The set-point of $A$ is 3.0. (a) Step-wise increase of $k_1$ from 1.0 to 5.0 at time $t$=100 at different and constant background perturbations $k_3$ (0–128.0, applied in phases 1 and 2). Note the successive decrease in the maximum excursion of $A$ ($\Delta A_{max}$) with slowed-down $A$ resetting kinetics as $k_3$ values increase. $\Delta A_{max}$ for $k_3$=4.0 is indicated. Rate constants (in au): $k_1$=1.0, $k_2$=0.0 (phases 1 and 2), $k_1$=5.0 (phase 2), $k_3$ variable, $k_4$=0.03, $k_5$=1.0, $k_6$=31.0, $k_7$=1×10$^{-6}$, $k_8$=0.1. Initial concentrations (in au): $A_0$=3.0, $E_0$=11.11 ($k_3$=0); $A_0$=3.0, $E_0$=22.22 ($k_3$=1); $A_0$=3.0, $E_0$=33.33 ($k_3$=2); $A_0$=3.0, $E_0$=55.55 ($k_3$=4); $A_0$=3.0, $E_0$=100.0 ($k_3$=8); $A_0$=3.0, $E_0$=188.89 ($k_3$=16); $A_0$=3.0, $E_0$=366.67 ($k_3$=32); $A_0$=3.0, $E_0$=722.22 ($k_3$=64); $A_0$=3.0, $E_0$=1433.33 ($k_3$=128). The inset shows the full adaptation response when $k_3$=128.0 (b) Relationship to Weber's law: When perturbation $k_1$ in phase 2 is adjusted such that the maximum (just noticeable) excursion $\Delta A_{max}$ is 0.03 (i.e. 1% of $A_{set}$) then both $k_1$ and the "perception" $E_{ss}$ are linear functions of the background $k_3$. Rate constants and initial concentrations as in (a), except that $k_1$ in phase 2 has the following values: **1**, $k_1$ = 1.0325 ($k_3$ = 2); **2**, $k_1$ = 1.0520 ($k_3$ = 4); **3**, $k_1$ = 1.0914 ($k_3$ = 8); **4**, $k_1$ = 1.1709 ($k_3$ = 16); **5**, $k_1$ = 1.3306 ($k_3$ = 32); **6**, $k_1$ = 1.6503 ($k_3$ = 64); **7**, $k_1$ = 2.2900 ($k_3$ = 128). (c) $\Delta A_{max}$ values as a function of background $k_3$ for three step perturbations in $k_1$. Note that the three curves are congruent, i.e., their identical shape can be precisely moved onto each other. (d) $t_{max}$ as a function of background $k_3$. For a given background $t_{max}$ is practically the same and independent of the three $k_1$ steps.

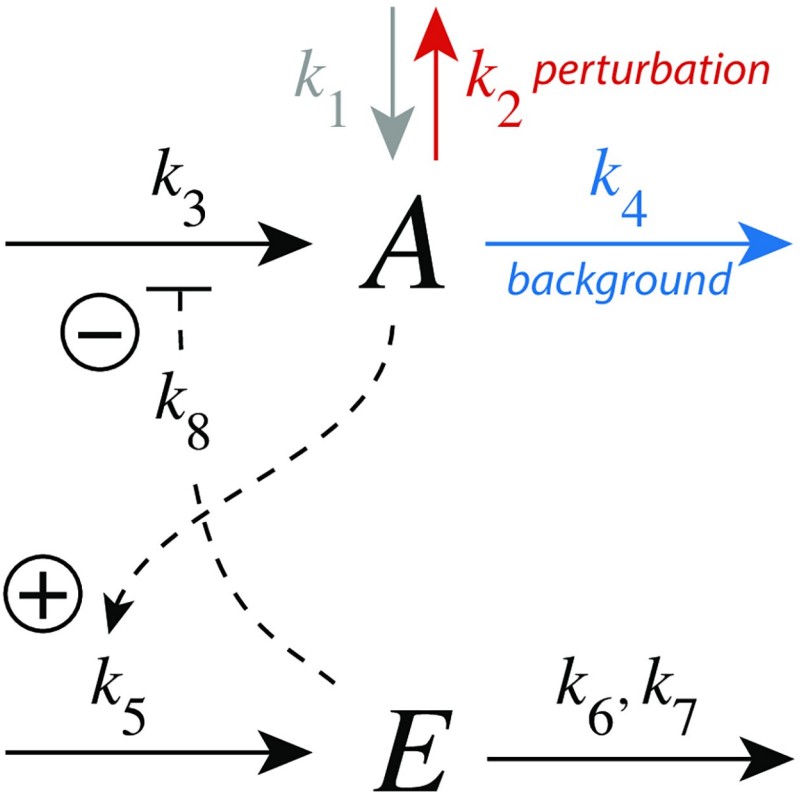

**Fig 7. Controller motif m2 with integral control implemented as a zero-order Michaelis-Menten (MM) type degradation of E.** Rate constant $k_2$ undergoes a step-wise change (perturbation), $k_3$ represents the maximum inflow of $A$, while $k_4$ is a (constant) background reaction. Rate constant $k_8$ is an inhibition constant. $k_6$ and $k_7$ are MM parameters analogous to $V_{max}$ and $K_M$, respectively. The grayed-out rate constant $k_1$ is set in the calculations to zero.

Fig 8a shows the response for step-wise changes in $k_2$ from 1.0 (phase 1) to 5.0 (phase 2) at different but constant background perturbations $k_4$. Typically for derepression controllers is both the decrease of $\Delta A_{max}$ at increasing backgrounds when a constant step perturbation is applied and a *decreasing* response time.

We were interested to see how the m2 controller would respond when a just noticeable excursion in $A$ ($\Delta A_{max}$) was applied for different background perturbations $k_4$. For that purpose we determined in phase 2 the steady state values of $E$ and the $k_2$ values when the excursion of $A$ was 1% of $A_{set}$(=3.0), i.e. $\Delta A_{max} = 0.03$. Fig 8b shows that ($1/E_{ss}$) and $k_2$ increase linearly with increasing $k_4$, a manifestation of Weber's law. In this view, ($1/E_{ss}$) could be interpreted as a "perceived" variable. Fig 8c and d show how $\Delta A_{max}$ and $t_{max}$ depend on the background $k_4$, respectively.

**Controller m2 with antithetic integral control.** Since we later will use bimolecular (antithetic) control [14, 19] to describe the simultaneous removal of $Ca^{2+}$ and $K^+$ out of a photoreceptor cell by potassium-dependent sodium-calcium exchangers (NCKX), we illustrate here how scheme m2 works with antithetic integral control (Fig 9).

The rate equations are:

$$\dot{A} = k_1 - k_2 \cdot A - k_4 \cdot A + \frac{k_3 k_8}{k_8 + E_1} \tag{13}$$

$$\dot{E}_1 = k_5 \cdot A - k_7 \cdot E_1 \cdot E_2 \tag{14}$$

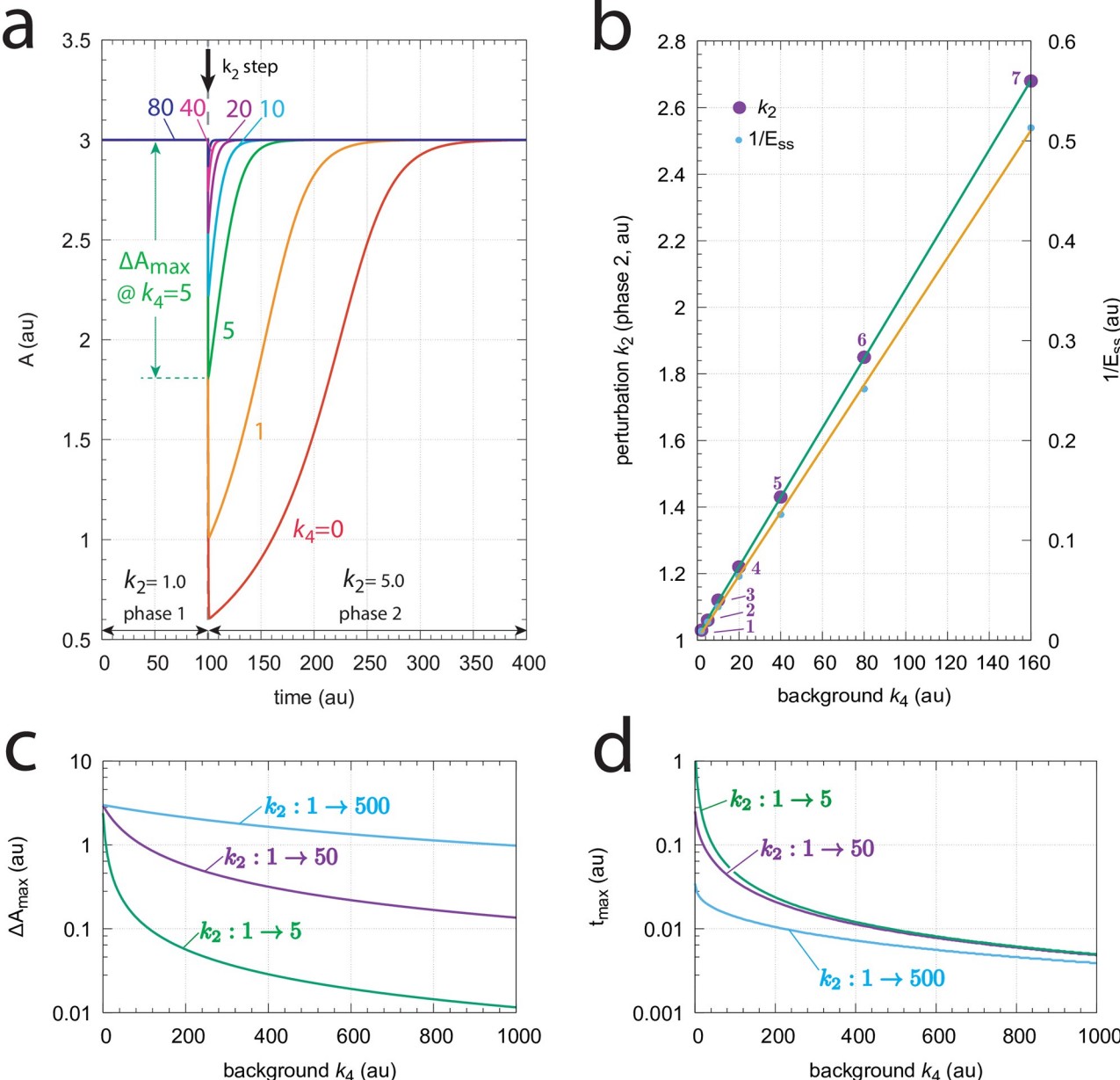

**Fig 8. Response kinetics and relationship to Weber's law in the m2 controller (Fig 7).** The set-point of $A$ is $A_{set}$=3.0 (a) Step-wise increase of $k_2$ from 1.0 to 5.0 at time $t$=100 at different and constant background perturbations $k_4$ (0–80). The maximum excusion in $A$, $\Delta A_{max}$, for $k_4$=5 is indicated. Note the successive decrease in $\Delta A_{max}$ and the more rapid resetting of $A$ at increased $k_4$ values. Rate constants (in au): $k_1$=0.0, $k_2$=1.0 (phase 1), $k_2$=5.0 (phase 2), $k_3$=1×10$^4$, $k_4$ variable, $k_5$=1.0, $k_6$=3.0, $k_7$=1×10$^{-6}$, $k_8$=0.1. Initial concentrations (in au): $A_0$=3.0, $E_0$=333.23 ($k_4$=0); $A_0$=3.0, $E_0$=166.62 ($k_4$=1); $A_0$=3.0, $E_0$=55.46 ($k_4$=5); $A_0$=3.0, $E_0$=30.20 ($k_4$=10); $A_0$=3.0, $E_0$=15.77 ($k_4$=20); $A_0$=3.0, $E_0$=8.03 ($k_4$=40); $A_0$=3.0, $E_0$=4.02 ($k_4$=80). (b) Relationship to Weber's law: in phase 2 the perturbation $k_2$ and ($1/E_{ss}$) are linear functions of the background perturbation $k_4$ when the "just noticable difference" $\Delta A_{max}$ is 0.03. Rate constants and initial concentrations as in (a), except that $k_2$ in phase 2 has the following values: **1**, $k_2$ = 1.0314 ($k_4$ = 2); **2**, $k_2$ = 1.0627 ($k_4$ = 5); **3**, $k_2$ = 1.1150 ($k_4$ = 10); **4**, $k_2$ = 1.2195 ($k_4$ = 20); **5**, $k_2$ = 1.4285 ($k_4$ = 40); **6**, $k_2$ = 1.8465 ($k_4$ = 80); **7**, $k_2$ = 2.6820 ($k_4$ = 160). (c) Monotonic decrease of $\Delta A_{max}$ as a function of background $k_4$ for three different steps. At constant background $\Delta A_{max}$ increases with increasing step size. (d) $t_{max}$ decreases monotonically with increasing backgrounds $k_4$. At constant background $t_{max}$ decreases with increasing step size. Rate constants in panels (c) and (d) are the same as for panel (a), apart from $k_2$ and $k_4$. Initial concentrations were taken as the steady state values of $A$ and $E$ at the different backgrounds $k_4$ prior to the applied step in $k_2$.

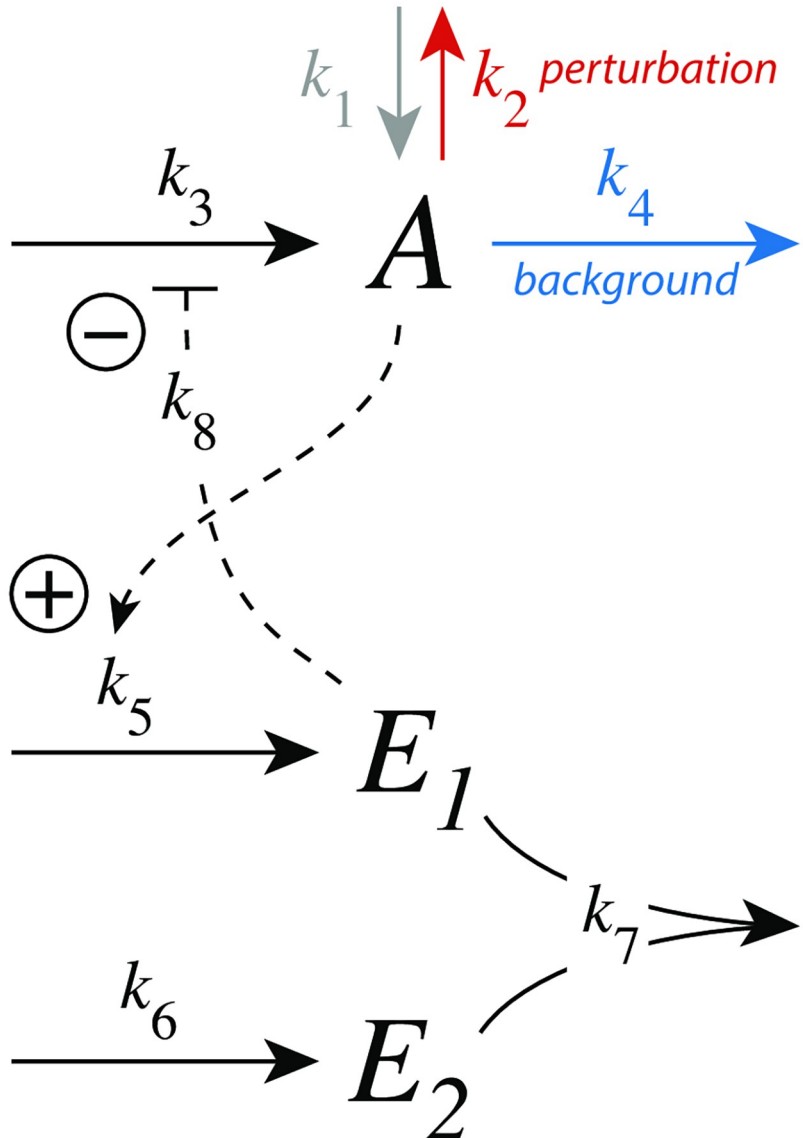

**Fig 9. Controller motif m2 with antithetic integral control.** Here, antithetic control is implemented as a bimolecular second-order reaction which removes the two controller molecules $E_1$ and $E_2$. See text on how $A$'s set-point is calculated.

$$\dot{E_2} = k_6 - k_7 \cdot E_1 \cdot E_2 \tag{15}$$

From the steady-state conditions for $E_1$ ($k_5 \cdot A_{ss}=k_7 \cdot E_1 \cdot E_2$) and $E_2$ ($k_6=k_7 \cdot E_1 \cdot E_2$) the set-point for $A$ ($A_{set}$) is given by:

$$k_5 \cdot A_{ss} = k_7 \cdot E_1 \cdot E_2 = k_6 \quad \Rightarrow \quad A_{ss} = A_{set} = \frac{k_6}{k_5} \tag{16}$$

In many respects robust perfect adaptation by zero-order or bimolecular (antithetic) kinetics, i.e., $E$ (Eq 12) and $E_1$ (Eq 14) behave dynamically identical. In fact, both $E$ and $E_1$ show zero-

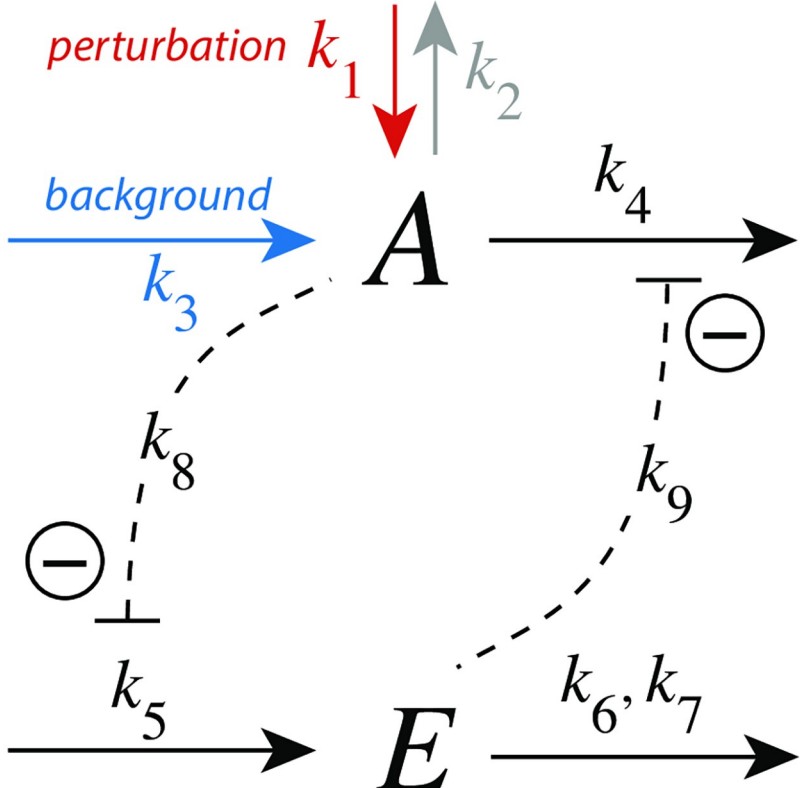

**Fig 10. Outflow controller motif m8 with integral control implemented as a zero-order Michaelis-Menten (MM) type degradation of E.** Rate constant $k_1$ undergoes a perturbation, while $k_3$ is a background inflow rate. $k_8$ and $k_9$ are inhibition constants. $k_6$ and $k_7$ are MM parameters analogous to $V_{max}$ and $K_M$, respectively. For simplicity, the grayed-out rate constant $k_2$ is set to zero.

order kinetics with respect to $E$ and $E_1$, respectively. In the supporting information 'S3 Text' we show the identical antithetic behavior of the m2 scheme when using step perturbations at various backgrounds in comparison with the above m2 calculations using zero-order kinetics.

**Controller m8.** Fig 10 shows the scheme of controller m8. The compensatory outflow flux $j_4 = k_4 \cdot k_9 \cdot A/((k_9 + E))$ and the signaling from $A$ to $E$ are based on derepression.

The rate equations are:

$$\dot{A} = k_1 - k_2 \cdot A + k_3 - k_4 \cdot A \cdot \left( \frac{k_9}{k_9 + E} \right) \tag{17}$$

$$\dot{E} = k_5 \cdot \left( \frac{k_8}{k_8 + A} \right) - \frac{k_6 \cdot E}{k_7 + E} \tag{18}$$

The set-point of $A$ is derived from the steady-state condition $\dot{E} = 0$ together with the assumption that $E$ is removed by zero-order kinetics, i.e. $E/(k_7 + E) \approx 1$:

$$\dot{E} = 0 \;\Rightarrow\; k_5 \cdot \left( \frac{k_8}{k_8 + A_{ss}} \right) = k_6 \;\Rightarrow\; A_{set} = A_{ss} = k_8 \left( \frac{k_5}{k_6} - 1 \right) \tag{19}$$

Fig 11a shows the response of the m8 derepression controller at different but constant backgrounds $k_3$. Note the typical, more rapid, resetting when backgrounds are increased. Panel b

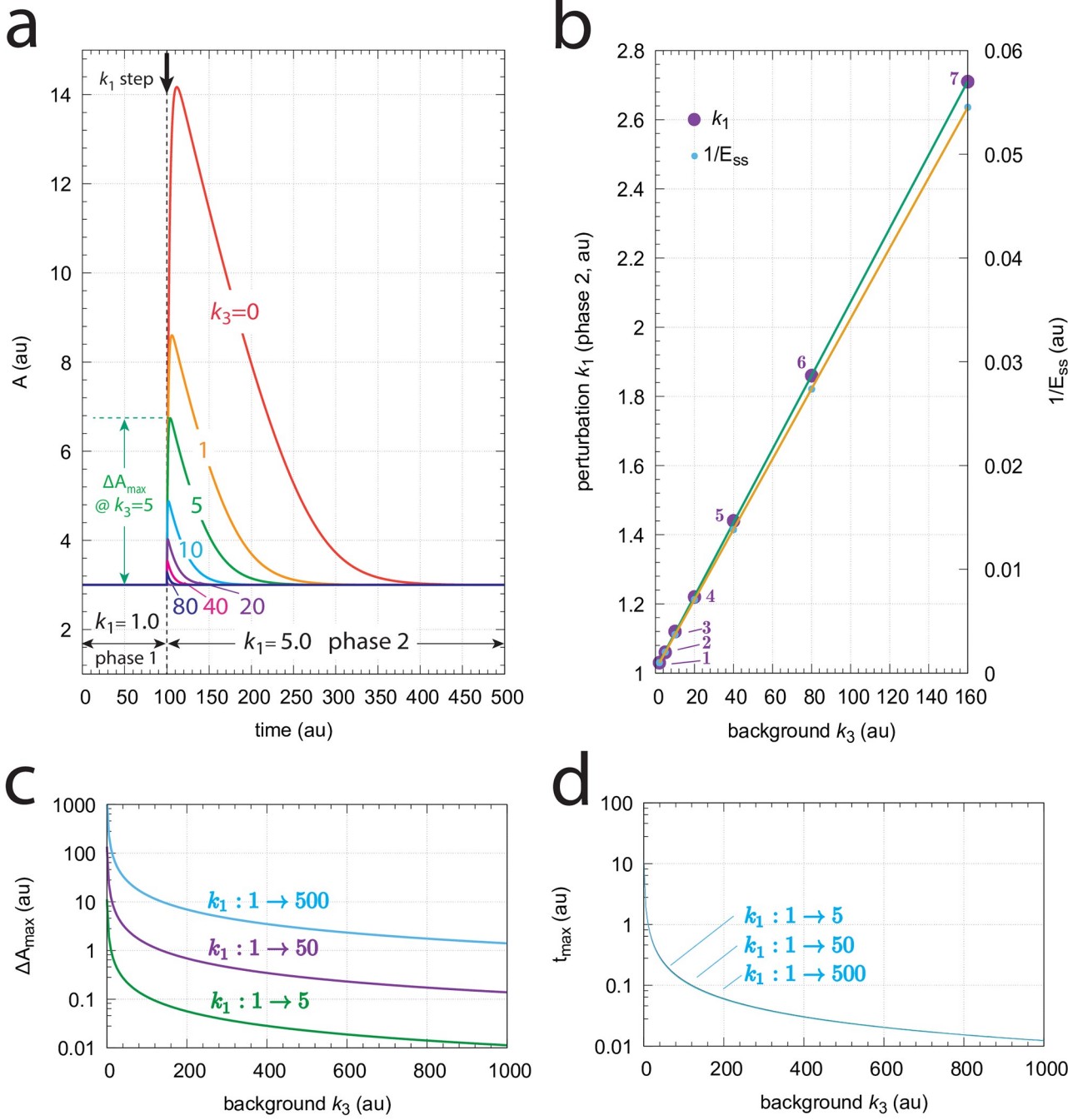

**Fig 11. Response kinetics and relationship to Weber's law in the m8 controller (Fig 10).** The set-point of $A$ is $A_{set}$=3.0. (a) Step-wise increase of $k_1$ from 1.0 to 5.0 at time $t$=100 at different and constant background perturbations $k_3$ (0–80). Note the successive decrease in the excursion of $A$ ($\Delta A_{max}$) and the more rapid $A$ resetting to the set-point at increased $k_3$ values. Rate constants (in au): $k_1$=1.0 (phase 1), $k_1$=5.0 (phase 2), $k_2$=0.0, $k_3$ variable, $k_4$ = $1\times10^4$, $k_5$=620.0, $k_6$=20.0, $k_7$=$1\times10^{-6}$, $k_8$=$k_9$=0.1. Initial concentrations (in au): $A_0$=3.0, $E_0$=2999.90 ($k_3$=0); $A_0$=3.0, $E_0$=1499.90 ($k_3$=1); $A_0$=3.0, $E_0$=499.90 ($k_3$=5); $A_0$=3.0, $E_0$=272.63 ($k_3$=10); $A_0$=3.0, $E_0$=142.76 ($k_3$=20); $A_0$=3.0, $E_0$=73.07 ($k_3$=40); $A_0$=3.0, $E_0$=36.94 ($k_3$=80). (b) Relationship to Weber's law: the perturbation $k_1$ and ($1/E_{ss}$) in phase 2 are linear functions of the background perturbation $k_3$ when $k_1$ is adjusted such that a "just noticable difference" of $\Delta A_{max}$=0.03 is observed. Rate constants and initial concentrations as in (a), except that $k_1$ in phase 2 has the following values: **1**, $k_1$ = 1.0319 ($k_3$ = 2); **2**, $k_1$ = 1.0637 ($k_3$ = 5); **3**, $k_1$ = 1.1169 ($k_3$ = 10); **4**, $k_1$ = 1.2231 ($k_3$ = 20); **5**, $k_1$ = 1.4356 ($k_3$ = 40); **6**, $k_1$ = 1.8604 ($k_3$ = 80); **7**, $k_1$ = 2.7111 ($k_3$ = 160). (c) $\Delta A_{max}$ values as a function of background $k_3$ for three step perturbations in $k_1$. Like for the m7 controller the three curves are congruent and their shape can be moved onto each other. (d) $t_{max}$ as a function of background $k_3$. For a given background $t_{max}$ values are practically the same independent of the three steps. Rate constants in panels (c) and (d) are the same as for panel (a), apart from $k_1$ and $k_3$. Initial concentrations are taken as the steady state values for $A$ and $E$ at the different backgrounds $k_3$ prior to the applied step in $k_1$.

shows that the controller follows Weber's law (Eq 1), i.e. when setting a "just noticeable difference" of $\Delta A_{max}$ to 1% of the set-point of $A$ ($A_{set}$=3.0) the required perturbations $k_1$ in phase 2 needed to achieve $\Delta A_{max}$=0.03 become a linear function of the background $k_3$. Similarly, plotting ($1/E_{ss}$) against the background is likewise linear, suggesting that ($1/E_{ss}$) may be interpreted as the "perception" of $\Delta A_{max}$. Fig 11c and 11d show how $\Delta A_{max}$ and $t_{max}$ depend on the background $k_3$, respectively.

## Implications to photoreceptor adaptation

As a biological example, we found a striking analogy between the resetting kinetics of the depression controllers m2, m4, m6, and m8 and the responses in vertebrate photoreceptors. In mammals and other animals photoadaptation occurs mainly in the retina, which consists of five basic classes of neurons: photoreceptors, bipolar cells, ganglion cells, horizontal cells, and amacrine cells, where each of them come in different subclasses. These neurons are arranged in layers and form a complex interaction network [21, 32, 33]. Our focus here is on the light-sensitive photoreceptor cells, which according to their physical shape are characterized as rods and cones, and differ in their sensitivity to light. Rods and cones occur in all retinas with the exception of the skate [33].

Fig 12 shows voltage responses of a rod cell to 10 ms light flashes at different background light intensities [34]. The experiments show that increased backgrounds lead to diminished

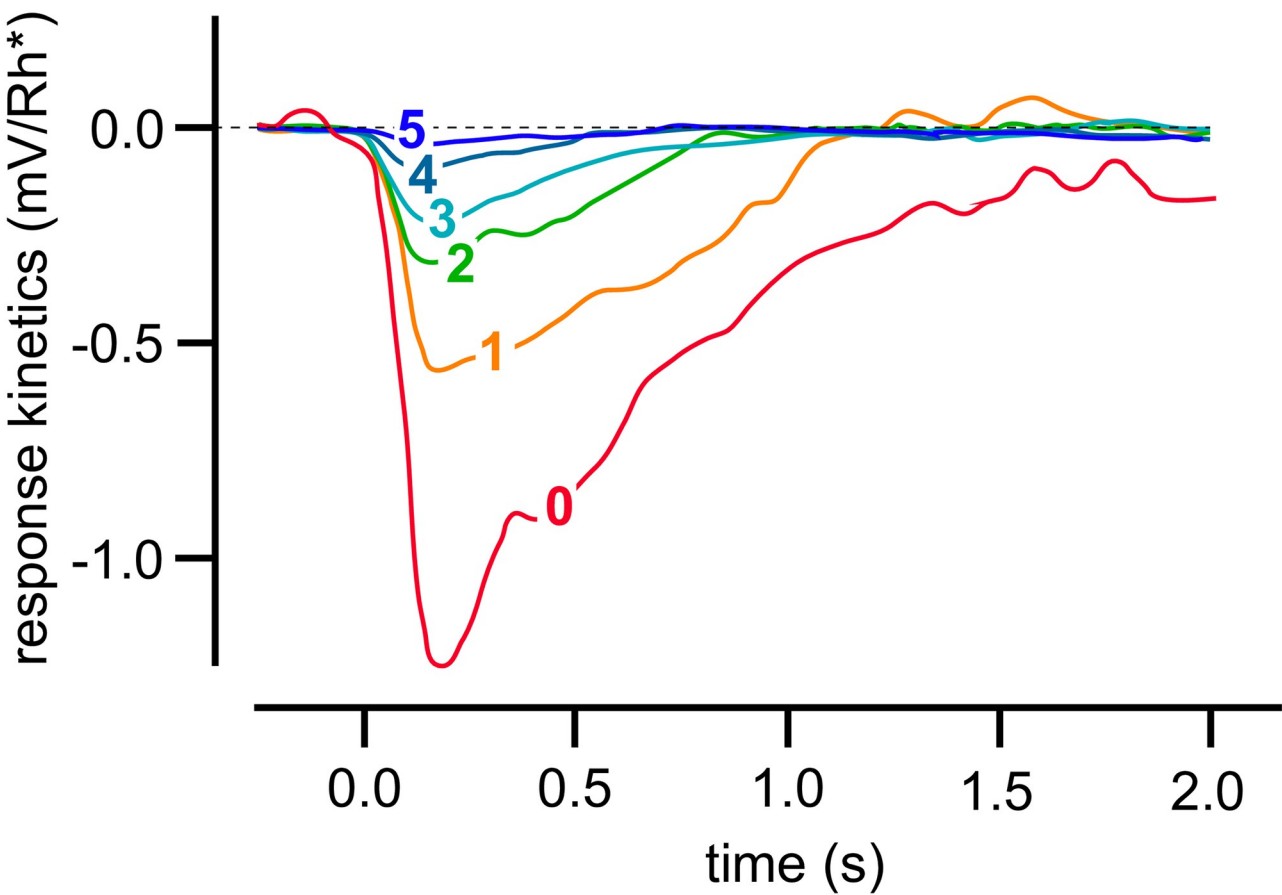

**Fig 12. Light adaptation in a Macaque monkey's rod cell.** 10 ms light flashes were applied to different light background intensities. Background intensities (in photons $\mu m^{-2} s^{-1}$) were: **0**, 0; **1**, 3.1; **2**, 12; **3**, 41; **4**, 84; **5**, 162. Redrawn after Fig 2A from Ref [34].

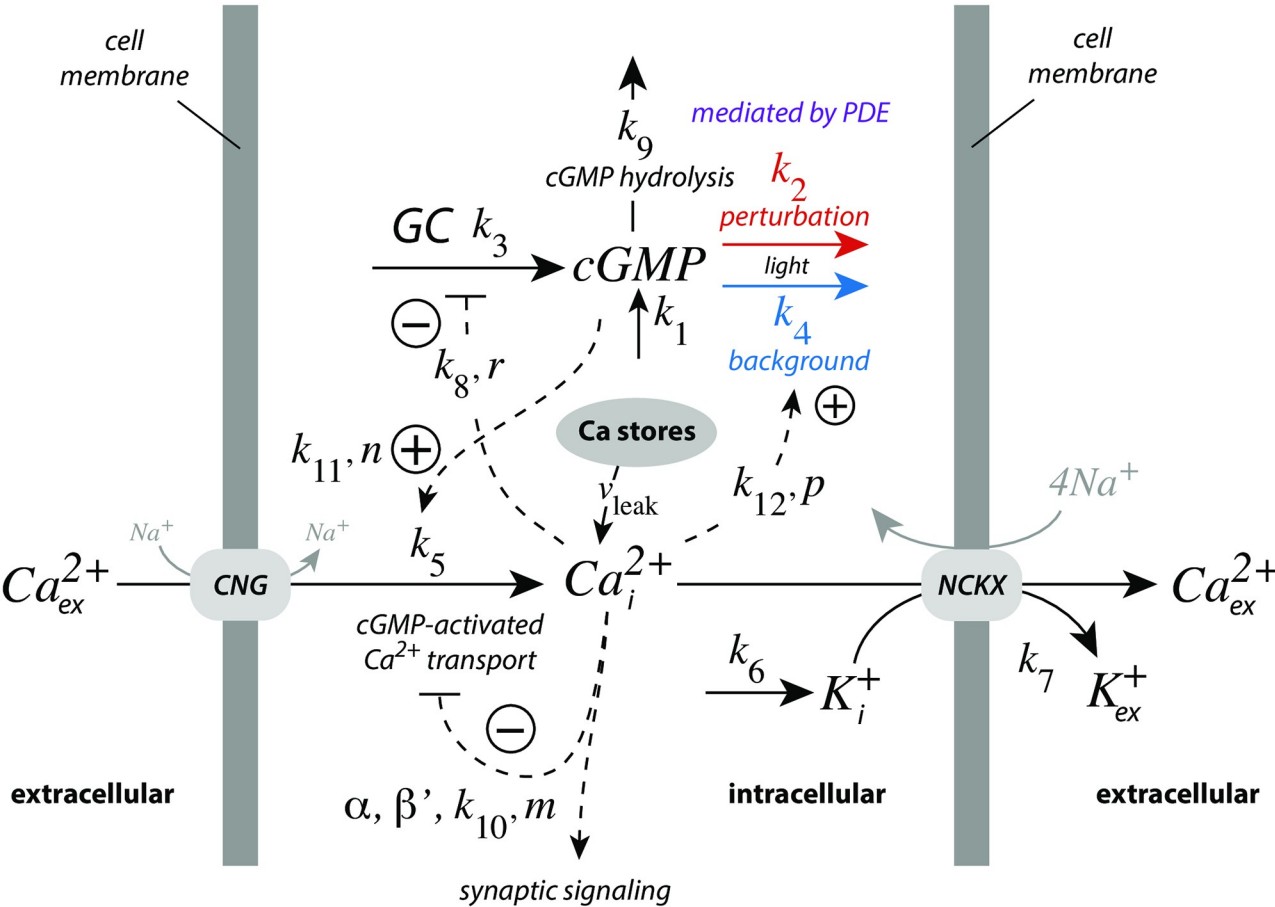

**Fig 13. Model with the main regulatory elements of vertebrate photoreceptor adaptation.** Light leads to the removal of cyclic guanosine monophosphate (cGMP) by phosphodiesterases (PDE), via transducin and the activation of PDE by internal $Ca^{2+}$ ($Ca_i^{2+}$). In the figure this path is split into two components, one background with rate constant $k_4$ (outlined in blue), and a perturbation on top of the background (rate constant $k_2$, outlined in red). cGMP is formed by guanylate cyclase (GC). cGMP's constitutive non-light induced hydrolysis is described by a first-order reaction with rate constant $k_9$. GC is inhibited/derepressed by $Ca_i^{2+}$ to keep cGMP under homeostatic control. cGMP activates cyclic nucleotide-gated (CNG) channels, which leads to the inflow of $Ca^{2+}$ into the cell, while high $Ca_i^{2+}$ levels inhibit CNG channels. Calcium is removed from the cell by potassium-dependent sodium-calcium exchangers (NCKX). Rate equations and used rate parameter values are described in the main text. Grayed reaction arrows indicate reactions which are not included in the model.

response excursions, while the resetting to the initial steady state levels were found to be faster. This behavior, a decreased sensitivity but accelerated response kinetics at increased background light intensities is considered typical for the light adaptation in vertebrate rod or cone cells [35]. When corresponding photocurrents are studied as a function of different background light levels the observed resetting behavior is close to that found for m8 or m6 controllers (for experimental data see Fig 1 in Ref [36]).

In photoreceptor cells cytosolic calcium has been found to be the major regulator in vertebrate light adaptation [37]. There, calcium takes part in a derepressing feedback loop analogous as $E$ in m2. Fig 13 shows a model with its main regulatory elements. In comparison with extracellular $Ca^{2+}$ concentrations, which are in the 10–100 mM range, cytosolic (internal) $Ca^{2+}$ levels ($Ca_i^{2+}$) are considerably lower, around in the 100 nM range since too high cytosolic $Ca_i^{2+}$ concentrations are toxic and may lead to apoptosis. While $Ca^{2+}$ is a versatile cellular signal its levels are also tightly regulated [38]. In photoreceptor cells dark $Ca_i^{2+}$ levels are in the

range around 300–500 nM [37], which is sufficient to regulate photo-transduction, but at the same time low enough to avoid cytotoxic $Ca^{2+}$ effects.

In vertebrate photoreceptor cells $Ca_i^{2+}$ is part of a negative feedback regulation of cyclic guanosine monophosphate (cGMP), where cGMP activates the inflow of $Ca^{2+}$ into the cytosol by cyclic nucleotide-gated (CNG) channels [37–39]. Analogous to a m2 controller, $Ca_i^{2+}$ on its side inhibits guanylate cyclase (GC), which synthesizes cGMP. In addition, $Ca^{2+}$ inhibits its inflow by CNG channels and takes part, analogous to a m5 controller, in the light-dependent removal of cGMP (with rate constants $k_2$ and $k_4$) by activating phosphodiesterases (PDE). Potassium-dependent sodium-calcium exchangers (NCKX) pump $Ca_i^{2+}$ out of the cell. In the model the removal of $Ca_i^{2+}$ by NCKX is formulated, for the sake of simplicity, as a bimolecular second-order reaction, where $K^+$ is removed together with $Ca_i^{2+}$, while keeping NCKX constant. For certain feedback combinations the bimolecular (or a zero-order) removal of $Ca_i^{2+}$ and $K^+$ by NCKX will lead to robust perfect adaptation of cGMP, which is discussed below. $k_1$ represents an inflow perturbation with respect to cGMP. We have mostly ignored $k_1$, except in section "Roles of the feedback loops", where $k_1$ is used to test the homeostatic behaviors of the individual feedback loops.

The rate equations of the model are:

$$\dot{\text{cGMP}} = k_1 + k_3 \left( \frac{k_8^r}{k_8^r + (Ca_i^{2+})^r} \right) - k_9 \cdot \text{cGMP} - \underbrace{(k_2 + k_4) \cdot (cGMP) \cdot \frac{(Ca_i^{2+})^p}{k_{12}^p + (Ca_i^{2+})^p}}_{light\ induced} \quad (20)$$

$$\dot{Ca_i^{2+}} = k_5 \cdot \frac{(cGMP)^n}{k_{11}^n + (cGMP)^n} \left( \alpha \cdot \frac{k_{10}^m}{k_{10}^m + (Ca_i^{2+})^m} + \beta' \right) - k_7(Ca_i^{2+})(K^+) + v_{\text{leak}} \quad (21)$$

$$\dot{K^+} = k_6 - k_7(Ca_i^{2+})(K^+) \quad (22)$$

**Estimation of model parameters.** Fig 14 gives an overview of the experimental data used to estimate some of the model parameters. Panel a shows the results by Koutalos et al. (Fig 3 in [40]; see also Fig 3 in [41]), who studied the influence of $Ca^{2+}$ on the light-stimulated PDE activity in salamander rods. The experimental data were described by the function

$$f(Ca_i^{2+}) = \frac{V_{max} \cdot (Ca_i^{2+})^p}{k_{12}^p + (Ca_i^{2+})^p} \quad (23)$$

with $V_{max}$=(100.01±2.53)%, $p$=0.894±0.0534, and $k_{12}$=(622.612±55.01)nM.

Also using salamander rods, Fig 14b shows the inhibition of GC activity by $Ca^{2+}$ when using 0.5 mM GTP (Fig 13 in [42]). The function

$$g(Ca_i^{2+}) = \frac{k_8^r}{k_8^r + (Ca_i^{2+})^r} \quad (24)$$

was fitted to the data with $k_8$=(57.49±2.53)nM and $r$=1.65±0.12.

Using bovine retinae, Hsu and Molday [39] determined the influence of cGMP and $Ca^{2+}$ on CNG channel activity in the presence of calmodulin (Fig 14, panels c and d, respectively).

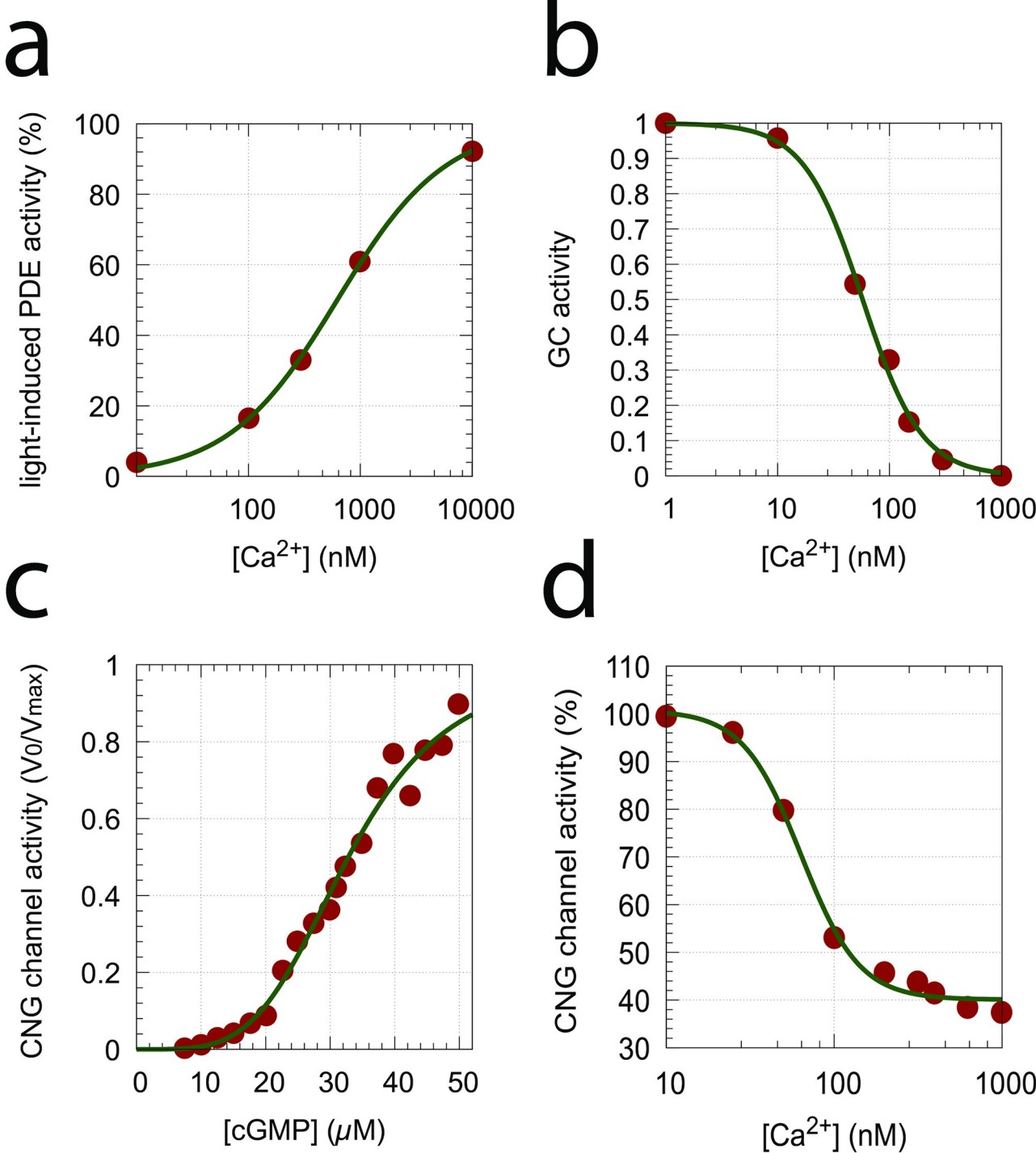

**Fig 14. Normalized experimental data used to extract parameter values.** (a) Light-induced PDE activity as a function of $Ca^{2+}$ concentration [40, 41]; (b) Inhibition of GC activity by $Ca^{2+}$ [42]; (c) CNG channel activity as a function of cGMP concentration [39]; (d) CNG channel activity as a function of $Ca^{2+}$ concentration [39].

For CNG channel activation by cGMP (panel c) the following trial function

$$h(cGMP) = \frac{(cGMP)^n}{k_{11}^n + (cGMP)^n} \tag{25}$$

described the experimental data with $k_{11}$=(32.81±0.39)$\mu$M and $n$=4.14±0.23 quite well. For the inhibition of the CNG channel by Ca$^{2+}$ (panel d) we fitted the function

$$k(\alpha, \beta, Ca_i^{2+}) = 100 \cdot \alpha \cdot \frac{k_{10}^m}{k_{10}^m + (Ca_i^{2+})^m} + \beta \tag{26}$$

to the experimental data obtaining $\alpha$=0.6067±0.0295, $k_{10}$=(63.57±4.44)nM, $m$=2.50±0.38, and $\beta$=40.07±1.29. In Eq 21 $\beta'$ is given by $\beta/100$.

Organelles, such as mitochondria and the endoplasmatic reticulum (ER), store calcium with relative high concentrations (100–800$\mu$M). There is evidence that intracellular Ca stores leak Ca into the cytosol [43–46]. Analyzing the data by Camello et al. [45] and Luik et al. [46], we observed (S4 Text and [47]) that the kinetics of the two recorded leaks were surprisingly different. While Camello et al. [45] found practically zero-order kinetics with respect to ER calcium and leak rates at around 0.25 $\mu$M/s, the data by Luik et al. [46] show clean *first-order* kinetics with respect to ER calcium. Here Ca-dependent leak velocities between 5.5 and 0.36 $\mu$M/s were observed (S4 Text). Also the results by Oldershaw et al. [43] and Missiaen et al. [44] indicate single or dual first-order kinetics in the decrease of store Ca. We wondered how calcium leaks may influence the photoadaptation of the model. As we will show in the section "Roles of the feedback loops" calcium leaks will have an influence on the steady state level of cGMP. In particular, when the leak rate $v_{leak}$ becomes larger than the K$^+$ inflow rate $k_6$ in the NCKX-based calcium pump, then uncontrolled growth in Ca$_i^{2+}$ may occur (S4 Text).

cGMP hydrolysis in darkness (rate constant $k_9$) is described as a first-order reaction with respect to cGMP. The value of $k_9$ is taken from the modeling work by Nikonov et al. (Table IV in [48]) with $k_9$=1.0s$^{-1}$. The rates for the light-induced removal of cGMP (described by $k_2$ and $k_4$) are variable (light-dependent) parameters.

Parameter $k_3$ represents the maximum rate of cGMP synthesis at low Ca$_i^{2+}$ concentrations. Its value ($k_3$=50 $\mu$M/s) has been taken from the work by Nikonov et al. [48].

The extrusion of Ca$_i^{2+}$ by NCKX is simplified as a second-order process with rate constant $k_7$, i.e. $v_{extrude} = k_7 \cdot (K^+) \cdot (Ca_i^{2+})$. Apart from that, we have not considered sodium ion and potassium ion currents.

It is interesting to note that in the absence of the CNG channel inhibition by Ca$_i^{2+}$ the NCKX pump would lead to robust perfect adaptation in cGMP by antithetic feedback [14], like the zero-order removal of $E$ in the above idealized controllers (see for example, Eq 12). However, such an antithetic control of cGMP without CNG channel inhibition by Ca$_i^{2+}$ would lead to high Ca$_i^{2+}$ concentrations and thereby to possible apoptosis of photoreceptor cells [49].

The remaining parameters $k_5$, $k_6$, and $k_7$ have been chosen such that cGMP and Ca$_i^{2+}$ levels are close to the observed experimental values [35, 37, 50, 51], i.e., using $k_5$=100 $\mu$M/s, $k_6$=0.5 $\mu$M/s, and $k_7$=2.0 $\mu$M$^{-1}$s$^{-1}$. While $k_7$ has no influence on the steady state values of cGMP and Ca$_i^{2+}$ it has a significant influence on how fast steady state levels are approached after light perturbations are applied (S5 Text).

**Application of pulse perturbations.** In the majority of experiments on rod or cone cells light perturbations are applied in form of flashes in the millisecond range (see for example Fig 12). Fig 15 shows the application of 10 ms pulses of light in the model. A $k_2$ pulse from $1 \rightarrow 50$ s$^{-1}$ is applied at time $t$=1.0 s for different $k_4$ backgrounds. In panel a the graphs are

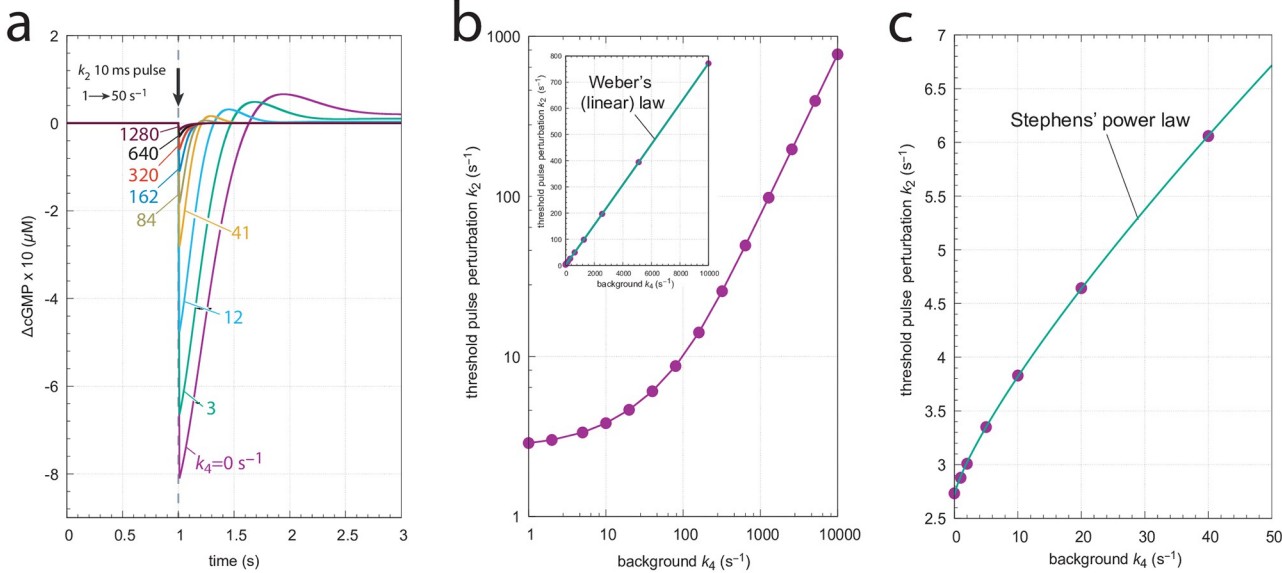

**Fig 15. Application of 10 ms $k_2$ pulses ($1 \rightarrow 50\ \text{s}^{-1}$) at different $k_4$ backgrounds.** (a) the scaled $\Delta$cGMP levels against time. Colored numbers indicate the different background levels in $\text{s}^{-1}$. Initial concentrations (in $\mu$M): cGMP$_0$=9.04191, $\text{Ca}_{i,0}^{2+} = 1.25717 \times 10^{-1}$, $\text{K}_0^+ = 1.25717$ ($k_4$=0 s$^{-1}$); cGMP$_0$=8.80039, $\text{Ca}_{i,0}^{2+} = 9.89550 \times 10^{-2}$, $\text{K}_0^+ = 2.52640$ ($k_4$=3 s$^{-1}$); cGMP$_0$=8.36375, $\text{Ca}_{i,0}^{2+} = 6.80242 \times 10^{-2}$, $\text{K}_0^+ = 3.67516$ ($k_4$=12 s$^{-1}$); cGMP$_0$=7.86039, $\text{Ca}_{i,0}^{2+} = 3.83237 \times 10^{-2}$, $\text{K}_0^+ = 6.52333$ ($k_4$=41 s$^{-1}$); cGMP$_0$=7.67946, $\text{Ca}_{i,0}^{2+} = 2.34638 \times 10^{-2}$, $\text{K}_0^+ = 1.06543 \times 10^1$ ($k_4$=84 s$^{-1}$); cGMP$_0$=7.61322, $\text{Ca}_{i,0}^{2+} = 1.31981 \times 10^{-2}$, $\text{K}_0^+ = 1.89421 \times 10^1$ ($k_4$=162 s$^{-1}$); cGMP$_0$=7.59537, $\text{Ca}_{i,0}^{2+} = 6.58562 \times 10^{-3}$, $\text{K}_0^+ = 3.79615 \times 10^1$ ($k_4$=320 s$^{-1}$); cGMP$_0$=7.59210, $\text{Ca}_{i,0}^{2+} = 3.09110 \times 10^{-3}$, $\text{K}_0^+ = 8.087735 \times 10^1$ ($k_4$=640 s$^{-1}$); cGMP$_0$=7.59160, $\text{Ca}_{i,0}^{2+} = 1.42894 \times 10^{-3}$, $\text{K}_0^+ = 1.74954 \times 10^2$ ($k_4$=1280 s$^{-1}$). Panel b shows the threshold perturbation $k_2$, which leads to a $\Delta$cGMP of 0.03 $\mu$M as a function of background. The overall curved log-log plot turns out to be linear and follows Weber's law (inset) as: threshold perturbation $k_2 = a \cdot (k_4)^n + b$ with a=(0.069±0.001)s$^{n-1}$, n=1.012±0.002, and b=(2.73±0.20)s$^{-1}$. Panel c shows that at low backgrounds the threshold-background relationship follows Stephens' power law, i.e., threshold perturbation $k_2 = a \cdot (k_4)^n + b$ with a=(0.175±0.006)s$^{n-1}$, n=0.800±0.009, and b=(2.72±0.01)s$^{-1}$. Parameter and rate constant values are as described in the previous section. See also 'S1 Programs' in S1 File.

scaled such that the steady state levels of cGMP are set to zero and the individual excursions in cGMP can be compared. As for the above derepression controllers m2, m4, m6 and m8 the excursion $\Delta$cGMP$_{\text{max}}$ of the controlled variable cGMP decreases with increasing backgrounds while the speed of resetting to its original steady state increases with increasing backgrounds (Fig 15a). These changes are considered to be typical for the light adaptation in vertebrate photoreceptors (for example, see Ch. V in [35] and Fig 22–19C in [21]).

Fig 15b shows threshold light pulse (10 ms) perturbations $k_2$ with a $\Delta$cGMP of 0.03 $\mu$M as a function of background light intensity $k_4$. The main graph shows the log-log plot which resembles the experimental results with rods or cones (see Fig 22–19B in [21]). The inset shows that the threshold-background relationship is linear in agreement with Weber's law, at least for large backgrounds. Panel c shows, on the other hand, that for small backgrounds the threshold-background relationship follows Stephens' power law. In fact, replotting the original experimental data [52] shown in Fig 22–19B of Ref [21], indicates that Stephens' law describes best the situation at low backgrounds, while at higher backgrounds the threshold-background relationship tends towards Weber's law (S6 Text).

**Application of step perturbations.** We applied step perturbations in the model to see to what extent the CNG channel inhibition by calcium affects cGMP homeostasis and avoids robust perfect adaptation. Fig 16a shows the influence of $k_{21} \rightarrow 50$ s$^{-1}$ steps at different backgrounds. The steps occur at time $t$=0.5 s and changes in cGMP are followed for 3 s. We also measured the maximum excursion of cGMP ($\Delta$cGMP$_{\text{max}}$) from its initial steady state level and the time $t_{\text{max}}$ at which $\Delta$cGMP$_{\text{max}}$ occurs (see inset).

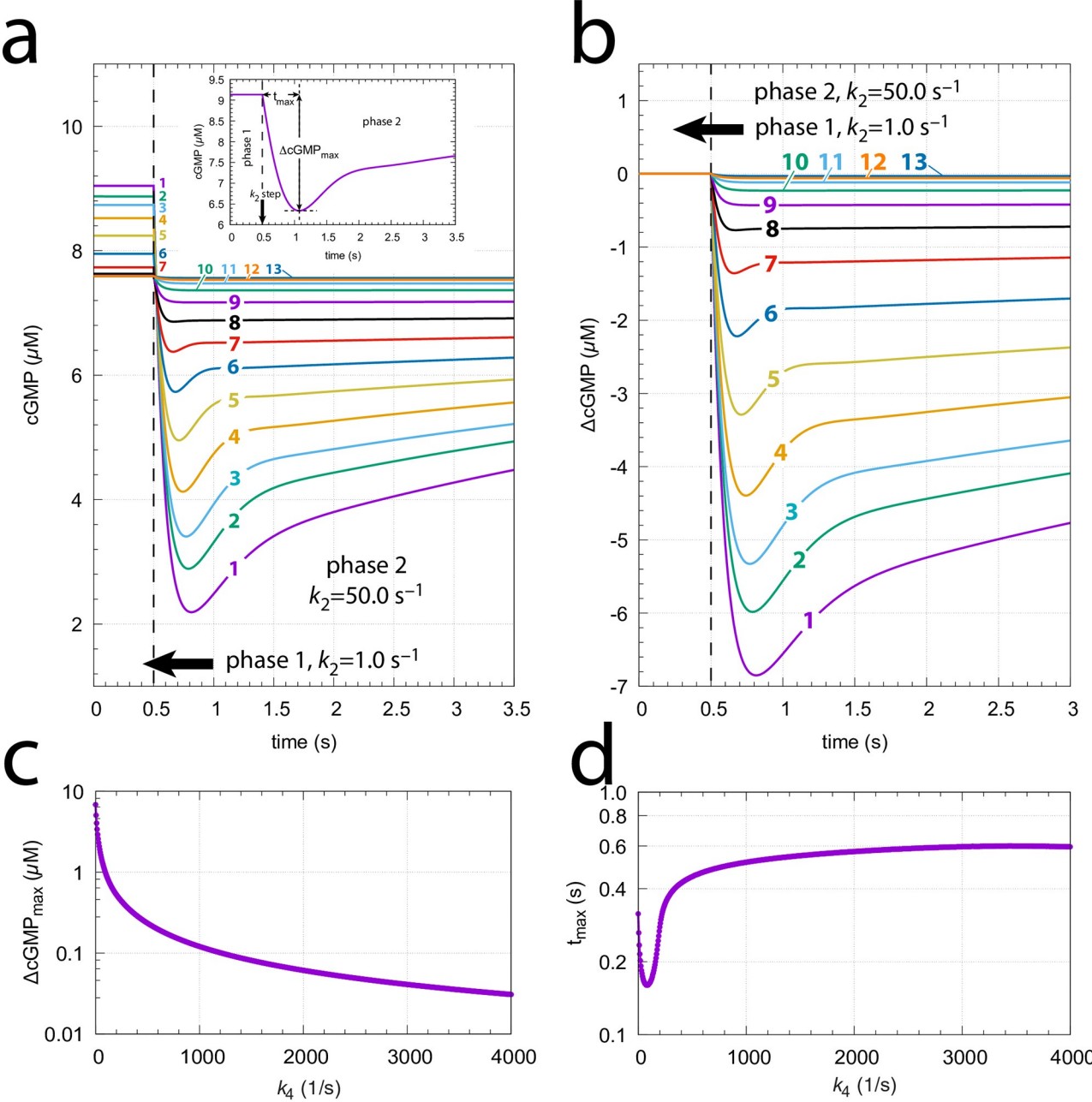

**Fig 16. The model's response towards $k_{21} \rightarrow$ 50 s$^{-1}$ steps at different backgrounds $k_4$.** (a) Unscaled cGMP concentrations as a function of time. The steps occur at $t$=0.5 s. Background $k_4$ values (s$^{-1}$): **1**, 0.0; **2**, 2.0; **3**, 4.0; **4**, 8.0; **5**, 16.0; **6**, 32.0; **7**, 64.0; **8**, 128.0; **9**, 256.0; **10**, 512.0; **11**, 1024.0; **12**, 2048.0; **13**, 4096.0. Initial concentrations (in $\mu$M): cGMP$_0$=9.04191, Ca$_{i,0}^{2+}$ = 1.25717, K$_0^+$ = 1.25717 ($k_4$=0 s$^{-1}$); cGMP$_0$=8.87243, Ca$_{i,0}^{2+}$ = 1.05733, K$_0^+$ = 2.36445 ($k_4$=2 s$^{-1}$); cGMP$_0$=8.73490, Ca$_{i,0}^{2+}$ = 9.33822, K$_0^+$ = 2.67717 ($k_4$=4 s$^{-1}$); cGMP$_0$=8.52196, Ca$_{i,0}^{2+}$ = 7.79015, K$_0^+$ = 3.20918 ($k_4$=8 s$^{-1}$); cGMP$_0$=8.24168, Ca$_{i,0}^{2+}$ = 6.08968, K$_0^+$ = 4.10531 ($k_4$=16 s$^{-1}$); cGMP$_0$=7.95044, Ca$_{i,0}^{2+}$ = 4.40458, K$_0^+$ = 5.67591 ($k_4$=32 s$^{-1}$); cGMP$_0$=7.73313, Ca$_{i,0}^{2+}$ = 2.87344, K$_0^+$ = 8.70036 ($k_4$=64 s$^{-1}$); cGMP$_0$=7.62877, Ca$_{i,0}^{2+}$ = 1.64393, K$_0^+$ = 1.52075 $\times 10^1$ ($k_4$=128 s$^{-1}$); cGMP$_0$=7.59845, Ca$_{i,0}^{2+}$ = 8.33466, K$_0^+$ = 2.99952 $\times 10^1$ ($k_4$=256 s$^{-1}$); cGMP$_0$=7.59259, Ca$_{i,0}^{2+}$ = 3.95393, K$_0^+$ = 6.3228 $\times 10^1$ ($k_4$=512 s$^{-1}$); cGMP$_0$=7.59165, Ca$_{i,0}^{2+}$ = 1.83312, K$_0^+$ = 1.36379 $\times 10^2$ ($k_4$=1024 s$^{-1}$); cGMP$_0$=7.59154, Ca$_{i,0}^{2+}$ = 8.44877, K$_0^+$ = 2.95901 $\times 10^2$ ($k_4$=2048 s$^{-1}$); cGMP$_0$=7.59152, Ca$_{i,0}^{2+}$ = 3.88973, K$_0^+$ = 6.42718 $\times 10^2$ ($k_4$=4096 s$^{-1}$). Inset: Defining $\Delta$cGMP$_{max}$ and $t_{max}$. (b) cGMP data as in (a), but scaled relative to their initial steady state concentrations. (c) and (d) $\Delta$cGMP$_{max}$ and $t_{max}$ values as a function of backgrounds $k_4$, respectively. Parameter and rate constant values are as described in section "Estimation of model parameters" (see also S1 Programs in S1 File).

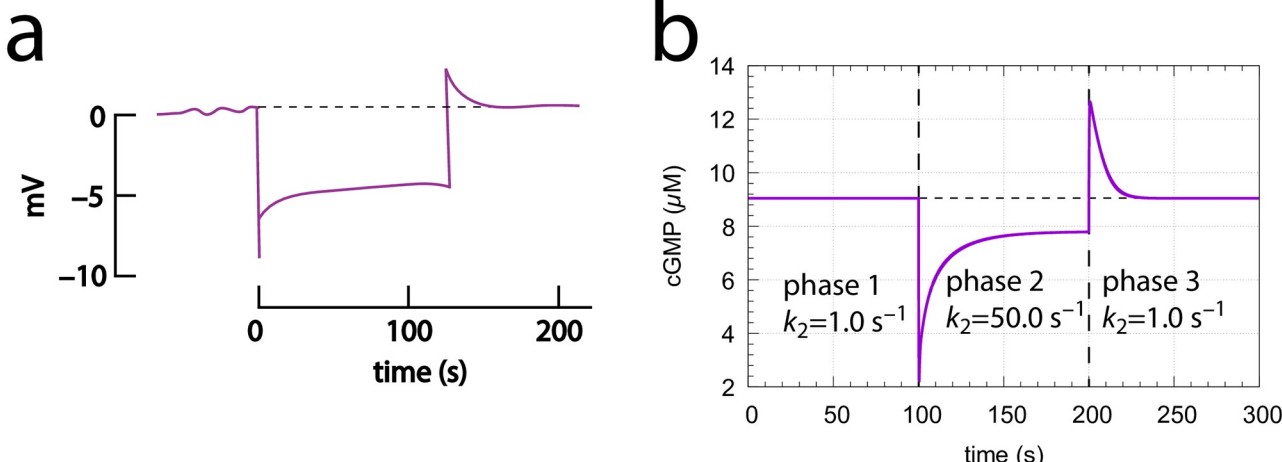

**Fig 17. Experimental and model behaviors when applying step perturbations.** (a) Experimental response of a red-sensitive turtle cone to a long step of light. Redrawn from Ref [53] (Fig 14, trace 2). (b) Model calculation using a $k_{21} \to 50 \ \text{s}^{-1}$ step at time $t$=100s. After 100 s $k_2$ returned to its original value. Background $k_4$=0.0 $\text{s}^{-1}$. All other rate parameters are as described in section "Estimation of model parameters". Initial concentrations: cGMP = 9.04$\mu$M, $\text{Ca}_i^{2+}$ = 125.7nM, $\text{K}^+$=2.0$\mu$M. See also S1 Programs in S1 File.

Fig 16b shows the same data as in (a), but scaled relative to their initial steady states. Due to the inhibition of CNG channels by calcium (Fig 13) the model does not show robust perfect adaptation (S5 Text, Fig 17). cGMP steady state levels during the step become significantly lower than their initial values before the step. This is seen in Fig 16a, where the pre-step steady state levels decrease as the background $k_4$ increases. Not unexpected we see that with increasing backgrounds the $\Delta \text{cGMP}_{\text{max}}$ excursions decrease monotonically (Fig 16c). Surprisingly, however, we find that $t_{\text{max}}$ first decreases, but then increases again (Fig 16d). Interestingly, when studying turtle photoreceptors, an increase of $t_{\text{max}}$ at increasing backgrounds has also been reported by Baylor and Hodgkin [53]. They studied both flashes and steps [54, 55] and provided several models [56] to explain the lengthening of the peak time $t_{\text{max}}$.

Fig 17a shows experimental results by Baylor and Hodgkin [53] when long steps of light are applied to red-sensitive turtle cones. The behavior of our model (panel b) is analogous with a typical overshooting when the step ends.

**Roles of the feedback loops.**   Outlined in Fig 18 are the three feedback loops in the model. Feedback loops 1 and 2, both based on the inflow activation of $\text{Ca}_i^{2+}$ by cGMP (outlined in purple), feed respectively back to cGMP by a $\text{Ca}_i^{2+}$-based inhibition (derepression) of cGMP synthesis (loop 1, analogous to m2, outlined in red) and by a $\text{Ca}_i^{2+}$-based activation of cGMP turnover (loop 2, analogous to m5, outlined in blue). Both loops 1 and 2 promote robust perfect cGMP homeostasis by antithetic control and oppose perturbations on cGMP. Feedback 3 (outlined in orange) keeps $\text{Ca}_i^{2+}$ levels low to avoid high and cytotoxic calcium levels inside the cell.

When feedback loop 3 is absent, for example by low $\text{Ca}_i^{2+}$ levels, the $\text{Ca}_i^{2+}$-inhibition term in Eq 21 becomes 1, because

$$\left( \alpha \cdot \frac{k_{10}^m}{k_{10}^m + (\text{Ca}_i^{2+})^m} + \beta' \right) \xrightarrow{\text{low Ca}_i^{2+}} \alpha + \beta' = 1 \qquad (27)$$

The remaining feedbacks 1 and 2 will provide robust perfect adaptation of cGMP, provided that there are sufficiently high GC and PDE activities to work as compensatory fluxes. This

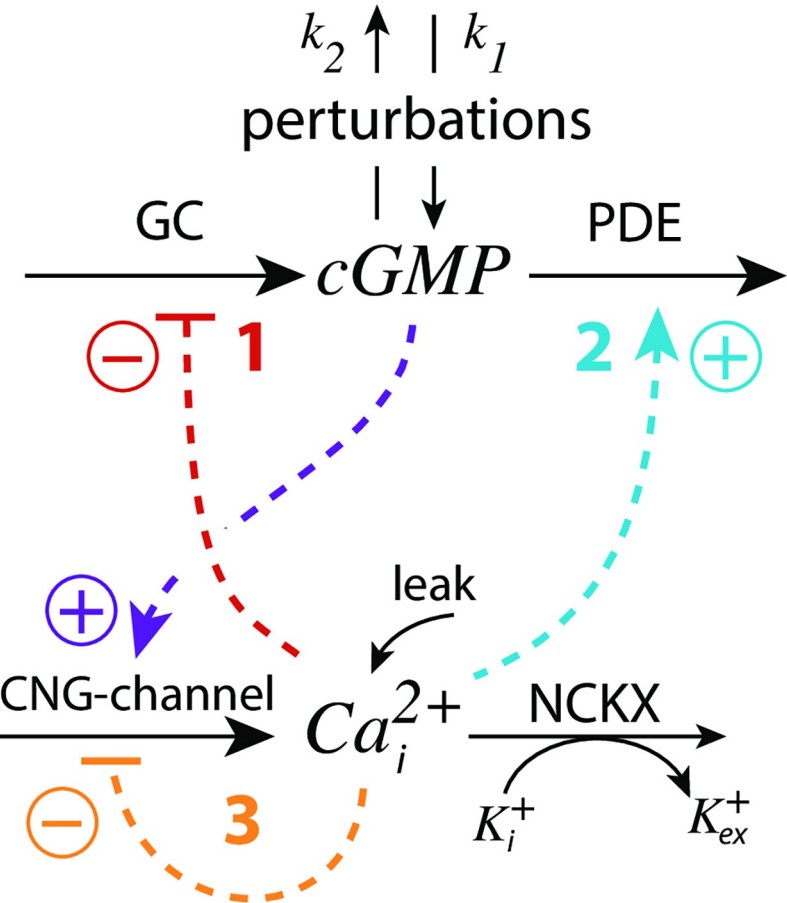

**Fig 18. Schematic outline of the feedback loops 1–3 in the model (Fig 13).** CNG: cyclic nucleotide-gated; GC: guanylate cyclase; PDE: phospho-diesterase; NCKX: potassium-dependent sodium-calcium exchangers (without the sodium part).

robust perfect adaptation in cGMP is due to the simultaneous NCKX-based removal of $Ca_i^{2+}$ and $K^+$ described by the term $k_7(Ca_i^{2+})(K^+)$ in Eqs 21 and 22. The $k_7(Ca_i^{2+})(K^+)$ transport term leads to robust antithetic integral control [14]. Instead of using the term $k_7(Ca_i^{2+})(K^+)$, one could have explicitly included the NCKX transporter protein, as generally outlined in [16] for catalyzed antithetic controllers. Anyway, using the $k_7(Ca_i^{2+})(K^+)$ term, the set-point of cGMP ($cGMP_{set}$) is calculated by setting Eqs 21 and 22 to zero and solving for cGMP. The resulting steady state concentration of cGMP becomes cGMP's set-point:

$$cGMP_{set} = cGMP_{ss} = k_{11}\sqrt[n]{\frac{b}{1-b}} \quad \text{with } b = \frac{k_6 - v_{\text{leak}}}{k_5} \tag{28}$$

Using the experimentally determined rate parameters (see section "Estimation of model parameters") leads to $cGMP_{set} = 7.61 \mu M$. The two feedback loops 1 and 2 act as an *antagonistic* pair as they will defend $cGMP_{set}$ robustly against both inflow and outflow perturbations, respectively. Fig 19a shows the homeostatic behavior of the loop 1–2 antagonistic feedback during three different phases where either inflow perturbation $k_1$ or outflow perturbation $k_2$ dominate. Although the antagonistic feedback can deal well with both inflow and outflow

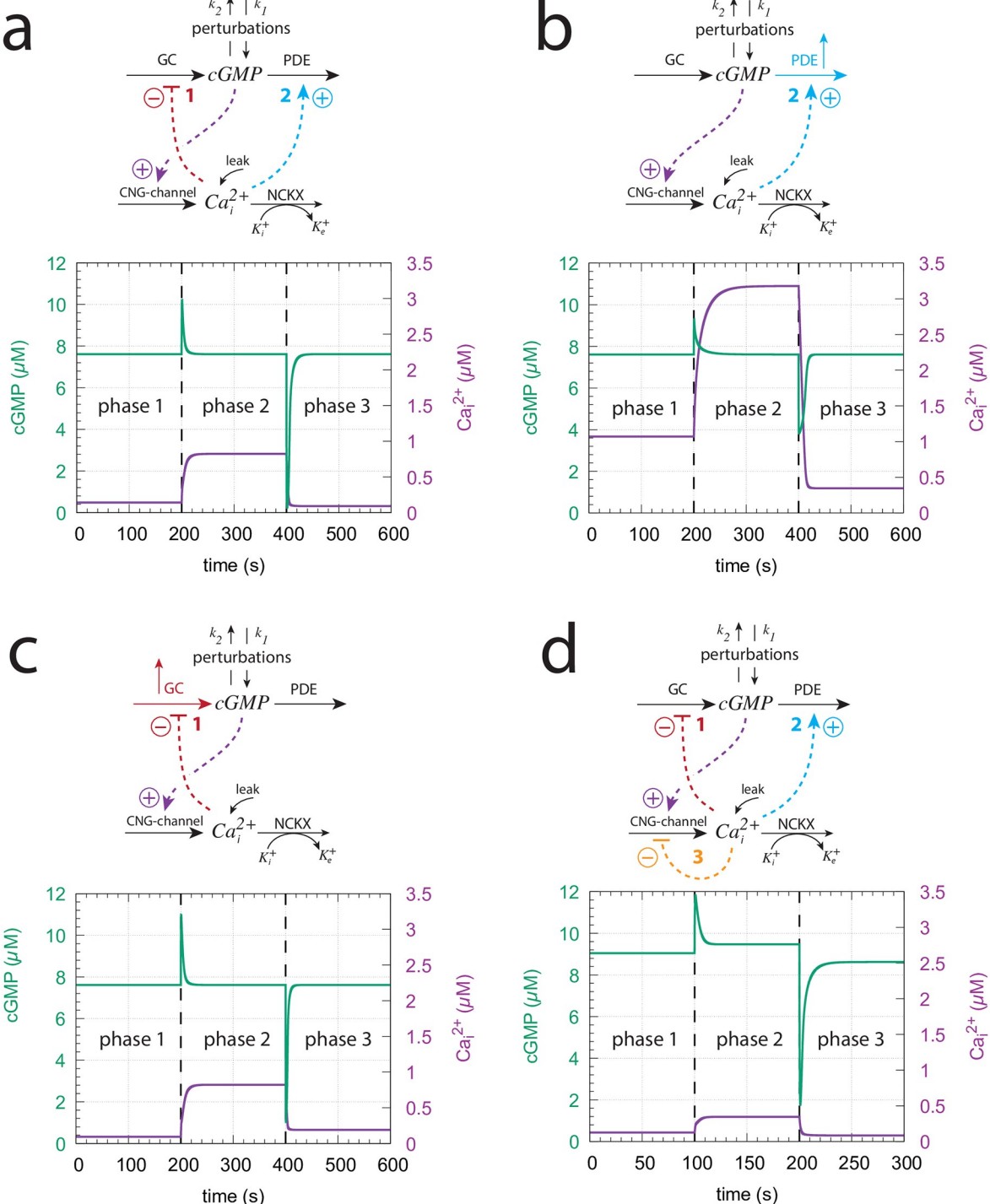

**Fig 19. Influence of the model's three feedback loops on the homeostatic behavior of cGMP and $Ca_i^{2+}$.** Perturbation profile in panels (a)-(d): phase 1: $k_1$=0.0$\mu$M/s, $k_2$=1.0s$^{-1}$, $k_4$=0.0s$^{-1}$; phase 2: $k_1$=7.0$\mu$M/s, $k_2$=0.0s$^{-1}$, $k_4$=0.0s$^{-1}$; phase 3: $k_1$=0.0$\mu$M/s, $k_2$=7.0s$^{-1}$, $k_4$=0.0s$^{-1}$. (a) Both feedback 1 and 2 are operative. Robust homeostasis of cGMP is observed with $cGMP_{set}$ = 7.61$\mu$M. Other rate constants values are as described in section "Estimation of model parameters". Initial concentrations: cGMP = 7.612$\mu$M, $Ca_i^{2+}$ = 141.7 nM, K$^+$=1.760$\mu$M. (b) Feedback 2 is only operative. In order to keep cGMP at its set-point $k_4$ needs to be increased to 8.0 $\mu$M/s in all three phases (indicated in the scheme by the blue upright arrow). Initial concentrations: cGMP = 7.612$\mu$M, $Ca_i^{2+}$ = 1.071 $\mu$M, K$^+$=2.335$\mu$M. (c) Feedback 1 is only operative. To keep cGMP at its set-point $k_3$ has been increased from 50.0 $\mu$M/s to 500.0 $\mu$M/s in phase 3 (indicated in the scheme by the red upright arrow). Initial concentrations: cGMP = 7.612$\mu$M, $Ca_i^{2+}$ = 94.9 nM, K$^+$=2.64$\mu$M. (d) All feedback loops are operative with rate constants as in panel (a). Although perfect adaptation in cGMP is lost both cGMP and $Ca_i^{2+}$ undergo only small variations when the perturbations are applied with lowest $Ca_i^{2+}$ levels. Initial concentrations: cGMP = 9.042$\mu$M, $Ca_i^{2+}$ = 125.7 nM, K$^+$=1.989$\mu$M. See S4 Text how the leak term affects this configuration.

perturbations it needs sufficiently large GC and PDE activities, reflected by sufficiently high $k_2$, $k_3$, and $k_4$ values, in order to provide the necessary compensatory fluxes.

Fig 19b shows the system's behavior when only feedback loop 2 is operative. To achieve control by only feedback 2 the condition in Eq 27 needs to hold and the inhibition of GC by $Ca_i^{2+}$ has to be abolished by using a high inhibition constant $k_8$. We have used $k_8 = 1 \times 10^9 \mu M$ with r = 1.0. When applying the same perturbation profile as in Fig 19a it turned out that the PDE activity from Fig 19a was not sufficient to keep cGMP homeostasis at $cGMP_{set} = 7.61 \mu M$. The reason for this is that the lack of feedback loop 1 causes a higher cGMP and $Ca^{2+}$ inflow into the cell. When becoming too high the $Ca^{2+}$ inflow cannot be absorbed by the constant $Ca_i^{2+}$ removal speed $k_6$ of NCKX. In other words, the antithetic zero-order removal kinetics of $Ca_i^{2+}$ by NCKX will become too slow and thereby lead to a steady increase (windup) in the concentration of $Ca_i^{2+}$ (S5 Text). To avoid this and to keep cGMP robustly at $cGMP_{set} = 7.61 \mu M$ we have in Fig 19b increased the background $k_4$ to 8 $\mu M/s$ (indicated by the blue upright arrow). Alternatively, one may increase the constant removal speed $k_6$ of the NCKX pump, but this will result in a change of $cGMP_{set}$ (see also S6 Text).

Fig 19c shows the system's behavior when only feedback loop 1 is present. To get only loop 1 operative the condition of Eq 27 is imposted and the activation constant $k_{12}$ (Fig 13) is set to zero. To act as a robust inflow controller cGMP homeostasis requires sufficiently high $k_3$ values. With the perturbation profile from panel (a) $k_3$ needs to be increased in phase 3 by one order of magnitude to $k_3 = 500 \mu/s$ (indicated by the red upright arrow in Fig 19c) in order to avoid cGMP levels below $cGMP_{set} = 7.61 \mu M$ (see also S6 Text).

When all three loops are operative (Fig 19d) the robust perfect adaptation of cGMP is lost due to the presence of feedback loop 3. However, with respect to the applied perturbations cGMP levels show only small variations and $Ca_i^{2+}$ steady state concentrations have their lowest values. The results in Fig 19 show that the antagonistic feedback between loops 1 and 2 is more efficient than when loops 1 or 2 are isolated. Although the robust perfect adaptation of cGMP is lost in the presence of feedback loop 3, the overlayed feedback structure between all three feedbacks provides a compromise between robust perfect adaptation of cGMP and the need to avoid high cytotoxic $Ca_i^{2+}$ levels.

Another aspect of the three feedbacks' overlay concerns the resetting times at varying/ increasing backgrounds. While a faster resetting with increasing backgrounds has been described as a typical property of vertebrate photoadaptation (see section V in [35]), in turtle photoreceptors Baylor et al. [53] found that increasing backgrounds first lead to a decrease in peak time (analogous to $t_{max}$), but further increases of the background eventually lead to an increase of the peak time ($t_{max}$), as qualitatively observed in Fig 16d. The increase of the time to peak was explained by Baylor et al. [56] by a hypothetical autocatalytic reaction which removed particles blocking the ionic channels. An additional factor could be a differential dominance between feedback loops 1 and 2, since loop 1 and loop 2 affect the resetting differently analogous as described for the m2 (Fig 8) and m5 (S1 Text) controllers.

Fig 20 shows ΔcGMP and $t_{max}$ as a function of the feedback arrangement. In panel (a) we have feedback loops 1 and 3 combined, while in panel (b) we have only feedback loop 2. When testing a $1.0 \to 50.0$ $\mu M/s$ $k_2$ step for increasing backgrounds both feedback arrangements show a monotonic decline of ΔcGMP as a function of background $k_4$ (middle panels), but differ in their $t_{max}$ responses (bottom panels). While combined feedback loop 1 and 3 show a monotonic shortening of $t_{max}$, in the feedback 2 arrangement $t_{max}$ first decreases, but then increases again as background $k_4$ increases, as found experimentally by Baylor et al. [53] and when all three feedback loops are combined (Fig 16c and 16d). Since the single feedback 2 behavior (Fig 20b) resembles that of all three feedbacks combined (Fig 19c and 19d) we

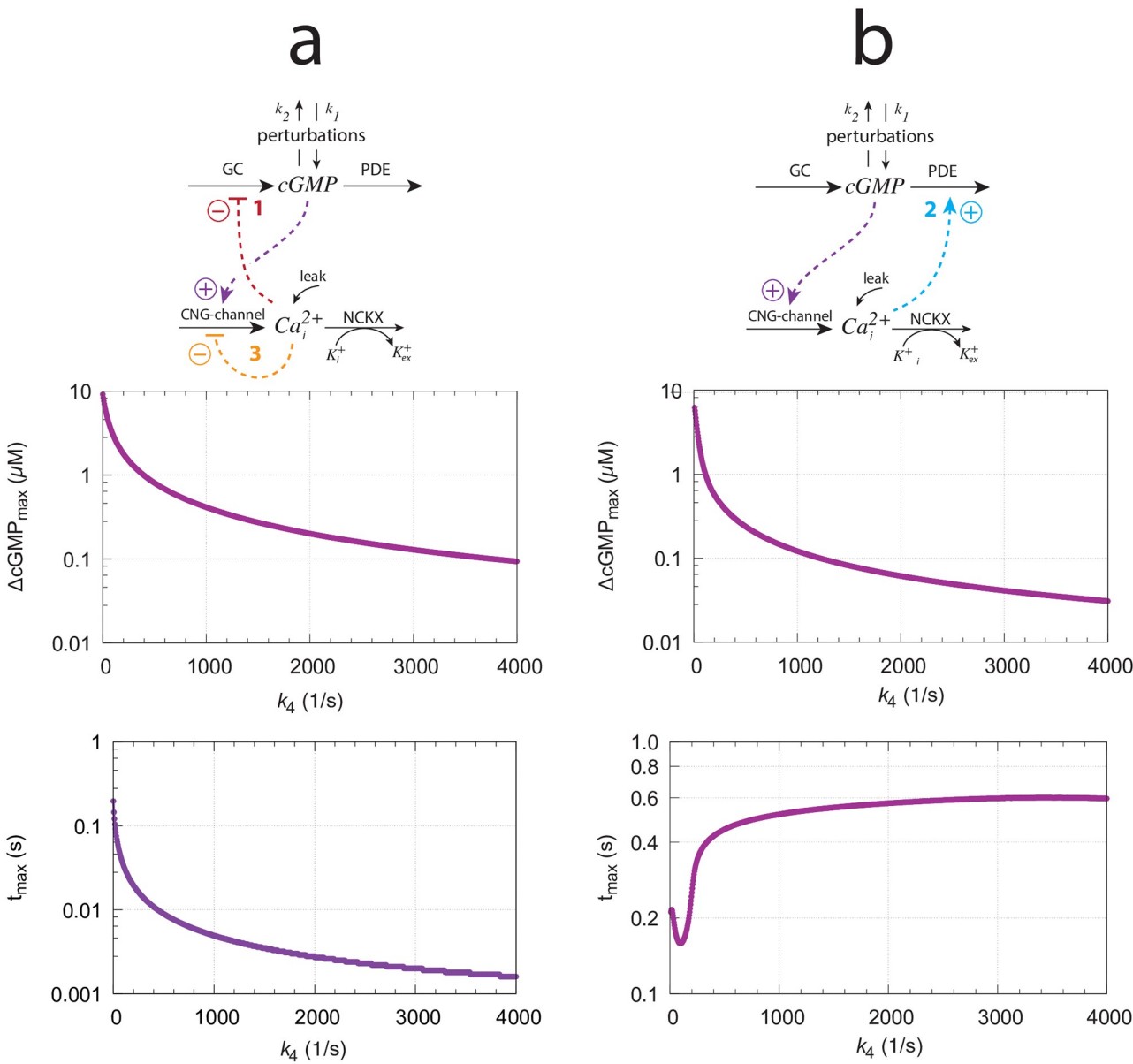

**Fig 20. The model's resetting behavior for different feedback arrangements when applying a $1.0 \rightarrow 50.0\ \mathrm{s^{-1}}$ step in $k_2$ as a function of backgrounds $k_4$.** (a) Feedback loops 1 and 3 are combined. (b) Feedback 2 only. Used parameter values, rate constants, and definition of $\Delta$cGMP and $t_{max}$ are as in Fig 16.

conclude that in our model with the used parameter values feedback 2 is dominating over the two other feedbacks with respect to the system's resetting behavior. In organisms where the photoadaptation shows faster resettings (decreasing or constant $t_{max}$) with increasing backgrounds, as found in Ref. [34] and highlighted in the review by Fain et al. [35], the feedback loop 2 may be weakened and loops 1 and 3 may become more dominant. Since the rate parameters of our model were taken from different organisms it is possible that these combined parameters reflect a situation closer to turtles [53] than, for example, to Macaque monkeys [34].

**Other model approaches.** There is an extensive literature in theoretical and computational approaches to understand various aspects of vertebrate photoadaptation. The approaches range from phenomenological mathematical descriptions to reaction kinetic and stochastic model calculations. For an overview we refer to chapter 19 in the book by Keener and Sneyd [57], to the review by Roberts et al. [58] and to Pan et al. [59]. Although phenomenological models can provide quantitative descriptions and predictions [60], they generally lack knowlegde of the involved chemical processes and their regulations. Due to this, the need for reaction kinetic descriptions has been emphasized [59, 61, 62].

Our approach, although primarily kinetic in nature, differs from previous adaptation models by looking at photoadaptation from a robust homeostatic viewpoint. In this respect we agree with Billman [63] that homeostatic approaches are still underappreciated and are far too often ignored as a central organizing principle in physiology.

## Conclusion and outlook

Studying perturbations with backgrounds on eight basic feedback loops m1-m8 with integral control show that these homeostatic controllers divide into two classes dependent on how the compensatory flux is activated. In the class where the compensatory flux is based on derepression faster resetting with respect to a standard step perturbation is observed when backgrounds increase. In the other class when compensatory fluxes are based on direct activation the resetting to the set-point slows down as backgrounds increase. In both cases the maximum excursion of the controlled variable following the perturbation decrease monotonically as backgrounds increase. We originally thought that vertebrate photoadaptation would be a nice example of using sole derepression kinetics in a robust control of cGMP with cellular calcium as the controller. However, the situations turned out to be more complex with an overlay of three feedback loops, one based on derepression by $Ca^{2+}$ on GC (feedback 1) and one based on $Ca^{2+}$-based light activation of PDE (feedback 2). The antagonistic pair of combined feedbacks 1 and 2 show more improved properties than each of the individual controllers alone. In addition, there is a third $Ca^{2+}$-controlling feedback (feedback 3) which apparently avoids high cytotoxic $Ca^{2+}$ levels. This combination of three feedback loops indicates that robust perfect adaptation of cGMP by feedback loops 1 and 2 is not by itself an evolutionary target, but that a compromise between these three controllers has developed by keeping both cGMP *and* cytosolic $Ca^{2+}$ levels at narrow limits, but not by robust perfect adaptation mechanisms. Furthermore, there is also evidence that photoadaptation with increasing backgrounds may both accelerate or slow down the resetting kinetics dependent on the dominance of feedback 1 or feedback 2.

The findings that controllers m1-m8 react so differently on perturbations with respect to backgrounds may be of importance also in other physiological systems. For example, blood sugar levels are controlled by two major feedback loops involving insulin and glucagon. Since glucose control by insulin is based by an activation of beta cells via glucose (see Supporting Material in Ref. [11]), constantly high glucose levels ("glucose overload") [64, 65], for example, may lead to a slower resetting of the insulin-based control loop in comparison with more rapid anticipated adaptations at lower glucose levels. Such a slowing-down response may be one of the causes that could participate in the mechanisms leading to insulin resistance and early diabetes. To what extent these aspects of background perturbations in homeostatic systems apply to the development of diabetes or have implications in other homeostatic systems needs certainly further investigations.

## Supporting information

**S1 File. Documentation.** (part 1). A zip-file with python scripts describing the results for motifs m1 (Fig 4a), m7 (Fig 6a), m2 (Fig 8a), m8 (Fig 11a), m3 and m5 (S1 Text, Figs S2a and S4a), m4 (S2 Text, Fig S2), and m6 (S2 Text, Fig S4a). (part 2). A zip-file with python scripts describing the results for Figs 15, 16a, 16b, 17b and 19.
(ZIP)

**S1 Text. Response kinetics of controllers m3 and m5.** Applied step perturbations lead to slower resetting kinetics for increasing backgrounds.
(ZIP)

**S2 Text. Response kinetics of controllers m4 and m6.** Applied step perturbations lead to faster resetting kinetics for increasing backgrounds.
(ZIP)

**S3 Text. Response kinetics controller m2 with antithetic integral control.** The behavior is dynamically identical to that of m2 with zero-order kinetics.
(ZIP)

**S4 Text. Influence of Ca leak kinetics on photoadaptation.** A comparison how experimentally observed zero-order and first-order Ca leak kinetics affect photoadaptation in the model and when homeostatic breakdown occurs.
(ZIP)

**S5 Text. Influence of $k_5$, $k_6$, and $k_7$ on the model's photoadaptation.** By using a $k_1$-$k_2$ perturbation profile influences of $k_5$, $k_6$, and $k_7$ on the model's resetting kinetics are shown.
(ZIP)

**S6 Text. Experimental light adaptation data.** Replots of experimental data show, as indicated by model calculations, that Stephens' law is followed at low backgrounds, while at higher backgrounds the response tends towards Weber's law.
(ZIP)

## Author Contributions

**Conceptualization:** Peter Ruoff.

**Formal analysis:** Jonas V. Grini, Melissa Nygård, Peter Ruoff.

**Investigation:** Jonas V. Grini, Melissa Nygård, Peter Ruoff.

**Methodology:** Peter Ruoff.

**Project administration:** Peter Ruoff.

**Software:** Jonas V. Grini, Melissa Nygård.

**Supervision:** Peter Ruoff.

**Validation:** Jonas V. Grini, Melissa Nygård.

**Visualization:** Peter Ruoff.

**Writing – original draft:** Peter Ruoff.

**Writing – review & editing:** Peter Ruoff.

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
