## [Decision Letter · Decision Letter 0]

27 Feb 2023

PONE-D-23-01979Homeostasis at different backgrounds: The roles of overlayed feedback structures in vertebrate photoadaptationPLOS ONE

Dear Dr. Ruoff,

Thank you for submitting your manuscript to PLOS ONE. After careful consideration, we feel that it has merit but does not fully meet PLOS ONE’s publication criteria as it currently stands. Therefore, we invite you to submit a revised version of the manuscript that addresses the points raised during the review process. It is fundamental to clarify both biological and computational aspects of this study (as highlighted by the reviewers) to make this work suitable for publication. I therefore suggested major revision.

We look forward to receiving your revised manuscript.

Kind regards,

Paolo Cazzaniga

Academic Editor

PLOS ONE

Journal Requirements:

**Additional Editor Comments:**

I agree with the reviewers that there are several aspects of this research study that must be clarified.

Reviewers' comments:

Reviewer's Responses to Questions

**Comments to the Author**

1. Is the manuscript technically sound, and do the data support the conclusions?

Reviewer #1: Yes

Reviewer #2: Yes

Reviewer #3: Yes

2. Has the statistical analysis been performed appropriately and rigorously? 

Reviewer #1: N/A

Reviewer #2: N/A

Reviewer #3: I Don't Know

3. Have the authors made all data underlying the findings in their manuscript fully available?

Reviewer #1: Yes

Reviewer #2: Yes

Reviewer #3: Yes

4. Is the manuscript presented in an intelligible fashion and written in standard English?

Reviewer #1: Yes

Reviewer #2: Yes

Reviewer #3: Yes

5. Review Comments to the Author

Reviewer #1: In line with their previous works on homeostatic controllers, Ruoff and colleagues address here the behaviour of basic controllers with respect to different but constant backgrounds. They first carefully analyse the dynamics of 8 motifs in response to a step perturbation at different backgrounds and show that these motifs can be divided into 2 classes, depending on their resetting timing, which is shown to be independent on weather they are inflow or outflow controllers. The dynamics of selected motifs is illustrated in the main paper while other motifs are described in the supplementary material. This gives a comprehensive overview of the response of the different motifs to perturbations (in terms of amplitude and speed, but also in relation to Weber law, which states that the perception of a "just noticeable" difference between a reference state and a slightly higher state is proportional to the reference state).

In the second part of the paper the authors focus on the photoreceptor system in vertebrates. This system contains 3 overlayed feedback loops whose the molecular components are well described. They involve calcium and cGMP, as well as the CNG channel and NCKX pump. After an estimation of the parameter values based on experimental observations, the authors study the adaptation of the system to pulse and to step perturbations. For the pulse perturbation, the authors found that the threshold (light stimulus needed to get a response of a given amplitude) - background relationship follows a Weber law for large background and a Stephen power law for small background, in agreement with the experimental observations. For the step perturbation, they found that the amplitude of the cGMP excursion decreases monotonically with the increase of the background (as expected) but that the response time first decreases and then increases with the background. This latter, non-intuitive prediction is related to observations done on turtle photoreceptors. Finally, the authors analyse in detail the role of the feedback loops. They found that 2 antagonistic feedbacks allow a robust adaptation while a third, negative feedback impairs the robustness of the adaptation but contributes to keep calcium level low (thus preventing high cytotoxic levels).

The paper is well structured and clearly written. The authors provide an elegant way to understand the kinetics of the vertebrate photoreceptor system and made clear links with experimental observations. The analysis of the various motifs provide a solid basis to understand the "logic" of the photoreceptor system but possibly also other systems. The codes to reproduce each figure are made available.

I was just wondering if other models exist for the photoreceptor system in vertebrates and how they differ from the present one, regarding the molecular components taken into account and/or the kinetics.

Reviewer #2: In the article entitled "Homeostasis at different backgrounds: the roles of overlayed feedback structures in vertebrate photoadaptation" by Grini et al., the authors classify the resetting behavior of eight possible basic integral regulatory motifs into two classes and discuss how the multiple feedback loops in molecular models of vertebrate photoadaptation are involved in controlling homeostasis. The work is original, has not been reported elsewhere, and the methods and analysis are of a high standard. The description is also adequate. However, some parts of the paper are confusing and there are several points that should be revised.

First, this reviewer's main concern is that this article should be divided into two separate studies, the study on the classification of basic integral controller motifs, and vertebrate photoadaptation molecular model dynamics. It is one outcome to discuss the existence of multiple feedback loops in the molecular model of vertebrate photoadaptation, each of which shows similarities to integral controller motifs, and their molecular dynamics. On the other hand, the finding that the possible basic integral controller motifs can be divided into two classes is also an achievement. Each of these is an independent achievement, but combining them into a single paper has blurred the point of contention. While it is possible to publish as is by addressing the following concerns, this reviewer believes that one option is to separate the independent results and publish them as two concise papers.

The reviewer was difficult to understand the meaning and significance of psychophysical laws (Weber's law and Stevens' power law). For example, there was a statement “Weber’s law is followed when a ”just noticeable difference” (threshold) of 1% of the controlled variable’s set-point was considered.” in the main text. What is the significance of following Weber's law? Similarly, the reviewer could not understand the significance of being able to express this in terms of Stevens' power law. These points could be explained in more detail for the unfamiliar reader.

The following points are minor comments.

p.2, Line 47: The constant C is not present in Eq 2.

p.2, Line 47: Isn't w the reference wight? Why the sudden change to general stimulus here?

p.3, Line 83: It should describe what the black arrow represents.

p.5, Line 140: Which one is Fig 4a panel (c) referring to? Perhaps there should be a comma after Fig 4a?

p.7, Line 170: The description of (Fig 1) after m2 and m8 is desirable.

p.8, Line 193: Major points have been made, but at least a brief mention should be made of Figs 8c and 8d.

p.8, Line 194: The reviewer believes this subsection should be moved to supporting information S3 Text, including Fig 9 because it is a little off the main subject.

p.9, Line 205: Eq 12, not Eq 11?

p.9, Line 222: This part should also be briefly explained at least for Fig 11 c and 11d.

p.11, Line 247: CNG channels are not familiar to the general reader. Appropriate references should be cited.

p.11, Line 249: No hyphen is required. Generally, "phosphodiesterase" is used.

p.12, 式21: This notation should correspond to the notation with vleak in Fig 13.

p.13, Line 310: Eq 11 wrong?

p.13, Lines 307-312: It is interesting … possibly toxic Ca2+ concentration. I feel that these sentences are off the main line. Is it really necessary?

Fig 14 and Fig 14 Legends: What is the unit of measure for the vertical axis in Fig 14b?

p.13, Line 314: Where did "experimental values" come from? Please cite the exact reference.

p.16, Lines 378-379: “Instead of … Ca2+ and K+ [16].” Does this sentence make sense? Shouldn't the extra information in the middle be removed because it makes it difficult to understand the main points?

Reviewer #3: The research here described is exclusively computational. No mention was made on any statistical evaluation of the results, but I do not know if this kind of simulation allows any since it is based on LSODA. a linear Solver for Ordinary Differential Equations. I do not feel adequate to judge if it is necessary.

6. PLOS authors have the option to publish the peer review history of their article (what does this mean?). If published, this will include your full peer review and any attached files.

Reviewer #1: No

Reviewer #2: **Yes: **Kunichika Tsumoto

Reviewer #3: **Yes: **Renata Tisi

---

## [Author Response · Author response to Decision Letter 0]

12 Mar 2023

please, see attached rebuttal letter and manuscript with track changes.

---

## [Editor Report · Decision Letter 1]

13 Mar 2023

PONE-D-23-01979R1Homeostasis at different backgrounds: The roles of overlayed feedback structures in vertebrate photoadaptationPLOS ONE

Dear Dr. Ruoff,

Thank you for submitting your manuscript to PLOS ONE. After careful consideration, we feel that it has merit but does not fully meet PLOS ONE’s publication criteria as it currently stands. Therefore, we invite you to submit a revised version of the manuscript that addresses the points raised during the review process.

In particular, the comments included in the review of Reviewer #3, sent in the previous communication as attached docx file, were not taken into account.

We look forward to receiving your revised manuscript.

Kind regards,

Paolo Cazzaniga

Academic Editor

PLOS ONE

Additional Editor Comments:

I am sending back the manuscript with major revision since the comments of Reviewer #3 included in the attached file were not taken into account during this round of revision.
---

## [Author Response · Author response to Decision Letter 1]

20 Mar 2023

Please, find attached file 'Response to reviewers'.

---

## [Decision Letter · Decision Letter 2]

30 Mar 2023

Homeostasis at different backgrounds: The roles of overlayed feedback structures in vertebrate photoadaptation

PONE-D-23-01979R2

Dear Dr. Ruoff,

We’re pleased to inform you that your manuscript has been judged scientifically suitable for publication and will be formally accepted for publication once it meets all outstanding technical requirements.

Kind regards,

Paolo Cazzaniga

Academic Editor

PLOS ONE

Additional Editor Comments (optional):

I suggest taking into account the minor modifications proposed by Reviewer 2.

Reviewers' comments:

Reviewer's Responses to Questions

**Comments to the Author**

1. If the authors have adequately addressed your comments raised in a previous round of review and you feel that this manuscript is now acceptable for publication, you may indicate that here to bypass the “Comments to the Author” section, enter your conflict of interest statement in the “Confidential to Editor” section, and submit your "Accept" recommendation.

Reviewer #1: All comments have been addressed

Reviewer #2: (No Response)

Reviewer #3: (No Response)

2. Is the manuscript technically sound, and do the data support the conclusions?

Reviewer #1: Yes

Reviewer #2: Yes

Reviewer #3: Yes

3. Has the statistical analysis been performed appropriately and rigorously? 

Reviewer #1: N/A

Reviewer #2: N/A

Reviewer #3: N/A

4. Have the authors made all data underlying the findings in their manuscript fully available?

Reviewer #1: Yes

Reviewer #2: Yes

Reviewer #3: Yes

5. Is the manuscript presented in an intelligible fashion and written in standard English?

Reviewer #1: Yes

Reviewer #2: Yes

Reviewer #3: Yes

6. Review Comments to the Author

Reviewer #1: (No Response)

Reviewer #2: The authors have nicely addressed all of my concerns and comments. However, the reviewer is still concerned with respect to psychophysical laws in this revision. As noted in the previous review comments, the reviewer could not find any statement regarding the physical significance (or biological significance with respect to the photoreceptor model) of showing Weber's law (or Stevens' power law). For example, on p. 15, Line 239, there is a sentence "When setting a ... a linear function of the background k3. What is the significance of the fact that it is a linear function of the background k3? Perhaps an unfamiliar reader like myself would like to know about that. I would appreciate any brief comments you can add.

Finally, on p. 2, line 42, there is a sentence "for example the by a human (or a receptor cell) perceived brightness" is it not "for example a human (or a receptor cell) perceived brightness"?

Reviewer #3: Although the authors did not accept some of the Reviewer's suggestions, they thoroughly discussed their point-of-view and justified their choices. In my opinion, the manuscript was largely improved during the revisions rounds and can now be published in this form.

7. PLOS authors have the option to publish the peer review history of their article (what does this mean?). If published, this will include your full peer review and any attached files.

Reviewer #1: No

Reviewer #2: **Yes: **Kunichika Tsumoto

Reviewer #3: **Yes: **Renata Tisi

---

## [Editor Report · Acceptance letter]

20 Apr 2023

PONE-D-23-01979R2 

Homeostasis at different backgrounds: The roles of overlayed feedback structures in vertebrate photoadaptation 

Dear Dr. Ruoff:

I'm pleased to inform you that your manuscript has been deemed suitable for publication in PLOS ONE. Congratulations! Your manuscript is now with our production department. 

Kind regards, 

on behalf of

Dr. Paolo Cazzaniga 

Academic Editor

PLOS ONE